# MATRIX-WISE CLASS IMBALANCE MATTERS: ON THE GENERALIZATION OF MICRO-AUC IN MULTI-LABEL LEARNING

## ABSTRACT

Micro-AUC is averaging AUC on the prediction matrix in multi-label learning. While it is a commonly-used evaluation measure in practice, the theoretical understanding is far behind. To fill up this gap, this paper takes an initial step to characterize the generalization guarantees of algorithms based on three surrogate losses w.r.t. Micro-AUC. Theoretically, we identify a critical data-dependent quantity affecting the generalization bounds: *the matrix-wise class imbalance*. Our results of the imbalance-aware error bounds show that the commonly-used univariate loss-based algorithm has a worse learning guarantee than the ones with the proposed pairwise and reweighting univariate loss, which probably implies its worse performance. Finally, empirical results of the linear and deep neural network-based models on various benchmarks corroborate our theory findings.

## 1 INTRODUCTION

As a typical learning task in machine learning, Multi-Label Learning (MLL) (McCallum, 1999) deals with settings where each instance can be associated with multiple labels. It has wide applications in various areas, e.g., computer vision (Carneiro et al., 2007), natural language processing (Schapire & Singer, 2000), and bioinformatics (Elisseeff & Weston, 2001). To comprehensively evaluate learning algorithms for MLL, various evaluation measures (Zhang & Zhou, 2014; Wu & Zhou, 2017) have been proposed from diverse perspectives, e.g., subset accuracy, Hamming loss, and ranking loss. Among them, Micro-AUC is a commonly-used measure in practice. Intuitively, Micro-AUC is averaging AUC on the prediction matrix, which is our focus in this paper.

Micro-AUC (and other measures) in MLL are discontinuous, non-differentiable, and non-convex, where optimizing them directly results in the NP-hard problem (Arora & Barak, 2009). Thus, one often seeks surrogate losses to construct learning algorithms for computational efficiency. Among them, perhaps the univariate surrogate loss (Boutell et al., 2004; Wu & Zhu, 2020) is the most commonly-used in practice, which originally aims to optimize Hamming loss. Empirically, many algorithms (including the commonly-used surrogate-based one) are often evaluated w.r.t. the Micro-AUC. Theoretically, however, the understanding is far behind. To fill up this gap, this paper attempts to answer the following questions formally:

(a) *What is the learning guarantee of the commonly-used surrogate univariate loss-based algorithms w.r.t. Micro-AUC?*

(b) *Can we design new surrogate losses to construct efficient and effective algorithms with better learning guarantees w.r.t. Micro-AUC?*

To answer the above questions, we propose an analytical framework to characterize the generalization guarantees of learning algorithms w.r.t. Micro-AUC. Inspired by the theory analyses, we propose one surrogate pairwise loss and another reweighted univariate loss. Formally, we analyze the learning guarantees of algorithms based on these three surrogates. Theoretically, we first identify a data-dependent quantity, i.e., the *matrix-wise class imbalance*, playing a critical role in these generalization bounds of algorithms.

Our results on the imbalance-aware bounds of these learning algorithms show that algorithms based on the proposed surrogate pairwise loss (i.e., $L_{pa}$) and reweighted univariate loss (i.e., $L_{u_2}$) can have better learning guarantees w.r.t. Micro-AUC than the one with the widely-used univariate loss (i.e., $L_{u_1}$) (see Table 1). Specifically, the pairwise loss-based algorithm $\mathcal{A}^{pa}$ has a learning guarantee of

Table 1: Summary of main theoretical results. The contributions of this paper are highlighted in red.

| Algorithm | Surrogate loss | Generalization bound | Computation |
|---|---|---|---|
| $\mathcal{A}^{pa}$ | pairwise ($L_{pa}$) | $\widehat{R}_S^{pa}(f) + O\left(\sqrt{\frac{1}{nK\tau_S}}\right)$ | $O(n^2K^2)$ |
| $\mathcal{A}^{u_1}$ (Boutell et al., 2004) | univariate ($L_{u_1}$) | $\frac{1}{\tau_S}\widehat{R}_S^{u_1}(f) + O\left(\frac{1}{\tau_S}\sqrt{\frac{1}{nK\tau_S}}\right)$ | $O(nK)$ |
| $\mathcal{A}^{u_2}$ | reweighted univariate ($L_{u_2}$) | $\widehat{R}_S^{u_2}(f) + O\left(\sqrt{\frac{1}{nK\tau_S}}\right)$ | $O(nK)$ |

$O\left(\sqrt{\frac{1}{nK\tau_S}}\right)$ w.r.t. Micro-AUC, where $n$ and $K$ are the numbers of instances and labels, respectively, and $\tau_S \in \left[\frac{1}{K}, \frac{1}{2}\right]$ denotes the matrix-wise class imbalance level of the training dataset $S$, where the smaller of $\tau_S$, the higher of the imbalance level. However, $\mathcal{A}^{pa}$ has a computational complexity of $O(n^2K^2)$, which is prohibitively expensive in large-scale problems. In comparison, the commonly-used univariate loss-based algorithm $\mathcal{A}^{u_1}$ has a learning guarantee of $O\left(\frac{1}{\tau_S}\sqrt{\frac{1}{nK\tau_S}}\right)$, which is worse than that of $\mathcal{A}^{pa}$. This implies that when the imbalance level of datasets is high (i.e., $\tau_S$ is small), $\mathcal{A}^{pa}$ would probably perform better than $\mathcal{A}^{u_2}$. Interestingly, the proposed reweighting univariate loss-based algorithm $\mathcal{A}^{u_2}$ has an error bound of $O\left(\sqrt{\frac{1}{nK\tau_S}}\right)$, which is the same as $\mathcal{A}^{pa}$. This also suggests that $\mathcal{A}^{u_2}$ probably has superior performance than $\mathcal{A}^{u_1}$. Computationally, $\mathcal{A}^{u_2}$ depends on $O(nK)$, which is the same as $\mathcal{A}^{u_1}$ and more efficient than $\mathcal{A}^{pa}$. Finally, our theory findings are corroborated by extensive experimental results of the linear and neural network-based models on various benchmark datasets.

Here we want to highlight the technical challenges for the generalization analysis of the Micro-AUC Maximization (MiAUCM) problem. Firstly, the formal definition of the true $(0/1)$ expected risk of Micro-AUC is more complex than the usual form in MLL, which we define for the first time to our knowledge. Secondly, the MiAUCM problem potentially involves learning with graph-dependent examples, which the existing techniques for the analysis of instance-based measures (Wu & Zhu, 2020; Wu et al., 2021) in MLL cannot handle, making it more challenging. Here we mainly follow the techniques in Bipartite Ranking (BR) (Usunier et al., 2005; Amini & Usunier, 2015).

Note that our theory is general for hypothesis space, where they can be various forms, e.g., linear, kernel, and deep neural network-based ones, which can be viewed to be orthogonal to the research of the complexity of hypothesis space (Bartlett et al., 2017; Long & Sedghi, 2019; Ma et al., 2020).

## 2 PRELIMINARIES

**Notations**. Let $\boldsymbol{a}$ and $\boldsymbol{A}$ denote a vector and matrix, respectively. Let $\mathbb{A}$ denote a set and $|\mathbb{A}|$ denote its cardinal number. $[n]$ denotes the set $\{1, 2, \ldots, n\}$. The indicator function $\mathbf{1}_{\text{condition}}$ returns 1 if the condition is true, 0 otherwise.

### 2.1 PROBLEM SETTING

Given a training sample $S = ((\boldsymbol{x}_1, \boldsymbol{y}_1), \ldots, (\boldsymbol{x}_n, \boldsymbol{y}_n))$ which is i.i.d. drawn from a distribution $P$, where $\boldsymbol{x}_i \in \mathcal{X}$, and $\boldsymbol{y}_i \in \{-1, +1\}^K$ for each $i \in [n]$. Besides, $y_{ij} = 1$ (or $y_{ij} = -1$) denotes that the instance $\boldsymbol{x}_i$ is relevant (or irrelevant) to the label $j$. The aim of MLL is to learn a good mapping function $h : \mathcal{X} \to \{-1, +1\}^K$.

A typical approach for MLL is to first learn a score function (or *predictor*) $f = (f_1, \ldots, f_K) : \mathcal{X} \to \mathbb{R}^K$ from some hypothesis space $\mathcal{F} = \{f\}$, and then get the final classifier through a thresholding function. Multi-Label Ranking (MLR) (Dembczynski et al., 2012) aims to learn a good predictor w.r.t. some ranking-based measure (e.g., Micro-AUC), which is our focus in this paper.

### 2.2 EVALUATION MEASURE

There are many evaluation measures for multi-label learning. Here we focus on one typical ranking-based measure, i.e., Micro-AUC, which is a micro-average of AUC on the prediction matrix. Given a training sample $S$ and a predictor $f$, the (empirical) Micro-AUC can be formally defined as

$$\textit{Micro-AUC}_S(f) = \frac{1}{|\mathbb{S}^+||\mathbb{S}^-|} \sum_{i=1}^{K} \sum_{j=1}^{K} \sum_{(p,q) \in \mathbb{S}_i^+ \times \mathbb{S}_j^-} \mathbf{1}_{f_i(\boldsymbol{x}_p) > f_j(\boldsymbol{x}_q)},$$

where $\mathbb{S}^+$ (or $\mathbb{S}^-$) denotes the relevant (or irrelevant) index pair set of (*instance*, *label*), and $\mathbb{S}_i^+$ (or $\mathbb{S}_j^-$) denotes the relevant (or irrelevant) instance index set, i.e.,

$$\mathbb{S}^+ \overset{\text{def}}{=} \{(p,i) \mid \forall(\boldsymbol{x}_p, \boldsymbol{y}_p) \in S, i \in [K], y_{pi} = +1\}, \quad \mathbb{S}_i^+ \overset{\text{def}}{=} \{p \mid \forall(\boldsymbol{x}_p, \boldsymbol{y}_p) \in S, y_{pi} = +1\},$$

$$\mathbb{S}^- \overset{\text{def}}{=} \{(q,j) \mid \forall(\boldsymbol{x}_q, \boldsymbol{y}_q) \in S, j \in [K], y_{qj} = -1\}, \quad \mathbb{S}_j^- \overset{\text{def}}{=} \{q \mid \forall(\boldsymbol{x}_q, \boldsymbol{y}_q) \in S, y_{qj} = -1\}.$$

Equivalently, maximizing the Micro-AUC is to minimize the following (empirical) risk (i.e., $1 - Micro\text{-}AUC_S(f)$):

$$\widehat{R}_S^{0/1}(f) = \frac{1}{|\mathbb{S}^+||\mathbb{S}^-|} \sum_{i=1}^{K} \sum_{j=1}^{K} \sum_{(p,q) \in \mathbb{S}_i^+ \times \mathbb{S}_j^-} L_{0/1}(\boldsymbol{x}_p, \boldsymbol{x}_q, f_i, f_j), \tag{1}$$

where the $0/1$ loss function $L_{0/1}(\boldsymbol{x}_p, \boldsymbol{x}_q, f_i, f_j) = \mathbf{1}_{f_i(\boldsymbol{x}_p) \leq f_j(\boldsymbol{x}_q)}$.

## 2.3 RISK

The true ($0/1$) expected (or generalization) risk w.r.t. Micro-AUC can be defined as

$$R_{0/1}(f) = \underset{i \sim P^+, j \sim P^-}{\mathbb{E}} \underset{\boldsymbol{x}_p \sim P_i^+, \boldsymbol{x}_q \sim P_j^-}{\mathbb{E}} \left[ L_{0/1}(\boldsymbol{x}_p, \boldsymbol{x}_q, f_i, f_j) \right], \tag{2}$$

where $\forall k \in [K]$, the (adjusted) distribution $P^+ \overset{\text{def}}{=} P^+(k) = \frac{P(y_k=1)}{\sum_{k \in [K]} P(y_k=1)}, P^- \overset{\text{def}}{=} P^-(k) = \frac{P(y_k=-1)}{\sum_{k \in [K]} P(y_k=-1)}$ for the label sampling, and the conditional distribution $P_k^+ \overset{\text{def}}{=} P(\boldsymbol{x}|y_k = 1), P_k^- \overset{\text{def}}{=} P(\boldsymbol{x}|y_k = -1)$ for the instance sampling. It can be verified that the empirical risk $\widehat{R}_S^{0/1}(f)$ in Eq.(1) is an unbiased estimator of the true expected risk, i.e., $\underset{S}{\mathbb{E}}\left[\widehat{R}_S^{0/1}(f)\right] = R_{0/1}(f)$.

The Micro-AUC (or $0/1$ loss) is discontinuous, non-differentiable, and non-convex, which results in the NP-hard problem (Arora & Barak, 2009). Thus, practically one often seeks (convex) surrogate losses to optimize it for computational efficiency. Given a surrogate loss function $L_\phi : \mathcal{X} \times \mathcal{X} \times \mathcal{F}_i \times \mathcal{F}_j \rightarrow \mathbb{R}_+$, where $i, j \in [K]$, $\mathcal{F}_k = \{f_k \mid f = (f_1, \ldots, f_K) \in \mathcal{F}\}$ for each $k \in [K]$ and we will discuss in detail in the next subsection, its empirical and expected risk can be defined as:

$$\widehat{R}_S^\phi(f) = \frac{1}{|\mathbb{S}^+||\mathbb{S}^-|} \sum_{i=1}^{K} \sum_{j=1}^{K} \sum_{(p,q) \in \mathbb{S}_i^+ \times \mathbb{S}_j^-} L_\phi(\boldsymbol{x}_p, \boldsymbol{x}_q, f_i, f_j), \qquad R_\phi(f) = \underset{S}{\mathbb{E}}\left[\widehat{R}_S^\phi(f)\right]. \tag{3}$$

Notably, we do not define the surrogate expected risk $R_\phi(f)$ as the following common form:

$$\underset{i \sim P^+, j \sim P^-}{\mathbb{E}} \underset{\boldsymbol{x}_p \sim P_i^+, \boldsymbol{x}_q \sim P_j^-}{\mathbb{E}} \left[ L_\phi(\boldsymbol{x}_p, \boldsymbol{x}_q, f_i, f_j) \right].$$

This is due to that the above form cannot cover the case of surrogate losses depending on the dataset $S$ while Eq.(3) can. Additionally, they are equal for certain losses that are independent of $S$.

## 3 LEARNING ALGORITHM

### 3.1 SURROGATE LOSS

Naturally, we propose the following surrogate pairwise loss to optimize Micro-AUC:

$$L_{pa}(\boldsymbol{x}_p, \boldsymbol{x}_q, f_i, f_j) = \ell(f_i(\boldsymbol{x}_p) - f_j(\boldsymbol{x}_q)), \tag{4}$$

where the base loss function $\ell(t)$ can be many commonly-used margin-based losses, e.g., the hinge loss $\ell(t) = \max(0, 1 - t)$ and logistic loss $\ell(t) = \log_2(1 + \exp(-t))$. As a natural property, $\ell(t)$ is an upper bound of the $0/1$ loss, i.e., $\mathbf{1}_{t \leq 0} \leq \ell(t)$. The empirical risk w.r.t. $L_{pa}$ is as follows:

$$\widehat{R}_S^{pa}(f) = \frac{1}{|\mathbb{S}^+||\mathbb{S}^-|} \sum_{i=1}^{K} \sum_{j=1}^{K} \sum_{(p,q) \in \mathbb{S}_i^+ \times \mathbb{S}_j^-} L_{pa}(\boldsymbol{x}_p, \boldsymbol{x}_q, f_i, f_j). \tag{5}$$

Perhaps, the most widely-used surrogate loss in MLL is the ordinary univariate loss $L_{u_1}$, which originally aims to optimize the Hamming loss measure. Its empirical risk can be written as

$$\widehat{R}_S^{u_1}(f) = \frac{1}{nK} \sum_{i=1}^{K} \sum_{a=1}^{n} \ell(y_{ai} f_i(\boldsymbol{x}_a)) \tag{6}$$

$$= \frac{1}{|\mathbb{S}^+||\mathbb{S}^-|} \sum_{i=1}^{K} \sum_{j=1}^{K} \sum_{(p,q) \in \mathbb{S}_i^+ \times \mathbb{S}_j^-} \left[ \frac{|\mathbb{S}^+|}{nK} \ell(f_i(\boldsymbol{x}_p)) + \frac{|\mathbb{S}^-|}{nK} \ell(-f_j(\boldsymbol{x}_q)) \right]. \tag{7}$$

Thus, we can define $L_{u_1}$ as the following form to optimize Micro-AUC:

$$L_{u_1}(\boldsymbol{x}_p, \boldsymbol{x}_q, f_i, f_j) = \frac{|\mathbb{S}^+|}{nK} \ell(f_i(\boldsymbol{x}_p)) + \frac{|\mathbb{S}^-|}{nK} \ell(-f_j(\boldsymbol{x}_q)). \tag{8}$$

Notably, $L_{u_1}$ cannot strictly upper bound the $0/1$ loss $L_{0/1}$, which is vital for its generalization guarantee w.r.t. the true $(0/1)$ risk.

To upper bound $0/1$ loss $L_{0/1}$, we propose another (reweighted) univariate loss as follows:

$$L_{u_2}(\boldsymbol{x}_p, \boldsymbol{x}_q, f_i, f_j) = \ell(f_i(\boldsymbol{x}_p)) + \ell(-f_j(\boldsymbol{x}_q)). \tag{9}$$

Its empirical risk can be written as

$$\widehat{R}_S^{u_2}(f) = \frac{1}{|\mathbb{S}^+||\mathbb{S}^-|} \sum_{i=1}^{K} \sum_{j=1}^{K} \sum_{(p,q) \in \mathbb{S}_i^+ \times \mathbb{S}_j^-} L_{u_2}(\boldsymbol{x}_p, \boldsymbol{x}_q, f_i, f_j). \tag{10}$$

Through a simple deduction, we can get

$$\widehat{R}_S^{u_2}(f) = \sum_{i=1}^{K} \sum_{a=1}^{n} \left[ \mathbf{1}_{a \in \mathbb{S}_i^+} \frac{1}{|\mathbb{S}^+|} \ell(f_i(\boldsymbol{x}_a)) + \mathbf{1}_{a \in \mathbb{S}_i^-} \frac{1}{|\mathbb{S}^-|} \ell(-f_i(\boldsymbol{x}_a)) \right]. \tag{11}$$

Interestingly, we can observe that, minimizing $\widehat{R}_S^{u_2}(f)$ has a computational efficiency over $\widehat{R}_S^{pa}(f)$. Computationally, $L_{u_2}$ can lead to a complexity of $O(nK)$ while $L_{pa}$ depends on $O(n^2K^2)$. Intuitively, compared with $L_{u_1}$, $L_{u_2}$ can be viewed as a reweighting strategy of losses to the relevant or irrelevant elements based on the label matrix, which could give more weight to the scarce ones.

### 3.2 LEARNING ALGORITHM

Here we consider learning algorithms using the Empirical Risk Minimization (ERM) rule (Shalev-Shwartz & Ben-David, 2014), where the constrained hypothesis space $\mathcal{F} = \{f = (f_1, \ldots, f_K)\}$ is the same. For these surrogate losses, their associated learning algorithms are as follows:

$$\mathcal{A}^{pa} : \min_{f \in \mathcal{F}} \widehat{R}_S^{pa}(f), \qquad \mathcal{A}^{u_j} : \min_{f \in \mathcal{F}} \widehat{R}_S^{u_j}(f), \ j = 1, 2. \tag{12}$$

Notably, our subsequent analyses are general, where $\mathcal{F}$ can be many common forms of hypothesis space, e.g., the linear, kernel, and neural network-based ones. Besides, these learning algorithms can often be equivalently transformed into the regularized ERM ones in practice.

## 4 MAIN RESULTS

In this section, we mainly give generalization analyses for the aforementioned learning algorithms, where we identify a data-dependent quantity of the *matrix-wise class imbalance* playing a vital role in these generalization error bounds. Besides, the detailed proof of the related lemmas, theorems, and corollaries are in Appendix B.

Firstly, we give the definition of the matrix-wise class imbalance w.r.t. a dataset.

**Definition 1** (**Matrix-wise class imbalance**). *For a dataset $S = ((\boldsymbol{x}_1, \boldsymbol{y}_1), \ldots, (\boldsymbol{x}_n, \boldsymbol{y}_n))$, define the following quantity to characterize its matrix-wise class imbalance level:*

$$\tau_S \stackrel{\text{def}}{=} \frac{\min\{|\mathbb{S}^+|, |\mathbb{S}^-|\}}{nK}, \quad \text{where } \tau_S \in \left[ \frac{1}{K}, \frac{1}{2} \right].[1]$$

Intuitively, $\tau_S$ reflects the class imbalance level of the label matrix of $S$, and that is why we name it that. The smaller of $\tau_S$, the higher of the matrix-wise class imbalance level. Besides, for the convenience of subsequent discussions, we give the following definitions.

**Definition 2** (**Matrix-wise class balanced or extremely imbalanced dataset**). *For a dataset $S$, it is said to be matrix-wise class balanced (or extremely class imbalanced) when $\tau_S = \frac{1}{2}$ (or $\tau_S = \frac{1}{K}$).*[2]

Next, we give the common assumption for the following analyses.

**Assumption 1** (**The common assumption**).

*(1) The training dataset $S = ((\boldsymbol{x}_1, \boldsymbol{y}_1), \ldots, (\boldsymbol{x}_n, \boldsymbol{y}_n))$ is an i.i.d. sample of the distribution $P$, where the input is bounded, i.e., $\forall \boldsymbol{x} \in \mathcal{X}$, $\|\boldsymbol{x}\|$ is upper bounded.*

*(2) The same hypothesis space $\mathcal{F} = \{f = (f_1, \ldots, f_K) : \mathcal{X} \to \mathbb{R}^K\}$ is constrained by some certain forms (e.g., the norm constraint).*

*(3) The base (convex) loss function $\ell(\cdot)$ is $\rho$-Lipschitz continuous and bounded by $B$.*

Since optimizing Micro-AUC potentially involves learning with graph-dependent examples, here we give the definition of the fractional Rademacher complexity of the loss and hypothesis spaces, which can handle the graph-dependent case.

**Definition 3** (**The fractional Rademacher complexity of the loss space and hypothesis space**). *Give the dataset $S = ((\boldsymbol{x}_1, \boldsymbol{y}_1), \ldots, (\boldsymbol{x}_n, \boldsymbol{y}_n))$, construct a dataset $\widetilde{S} = \{(\tilde{\boldsymbol{x}}_b, 1)\}_{b=1}^m = \left\{((\tilde{\boldsymbol{x}}_b^+, \tilde{\boldsymbol{x}}_b^-, \alpha_b, \beta_b), 1)\right\}_{b=1}^m$, where $\tilde{\boldsymbol{x}}_b^+ = \boldsymbol{x}_p$, $\tilde{\boldsymbol{x}}_b^- = \boldsymbol{x}_q$ for some $p, q \in [n]$, and $\alpha_b, \beta_b \in [K]$, $y_{p\alpha_b} = 1$, and $y_{q\beta_b} = -1$. Let $\{(I_j, \omega_j)\}_{j \in [J]}$ be a fractional independent vertex cover of the dependence graph $G$ constructed over $\widetilde{S}$ with $\sum_{j \in [J]} \omega_j = \chi_f(G)$, where $\chi_f(G)$ is the fractional chromatic number of $G$. For the hypothesis space $\mathcal{F} = \{f = (f_1, \ldots, f_K) : \mathcal{X} \to \mathbb{R}^K\}$ and the loss function $L : \mathcal{X} \times \mathcal{X} \times \mathcal{F}_i \times \mathcal{F}_j \to \mathbb{R}_+$, where $i, j \in [K]$, and $\mathcal{F}_k = \{f_k \mid f = (f_1, \ldots, f_K) \in \mathcal{F}\}$, $\forall k \in [K]$, the empirical fractional Rademacher complexity of the loss space w.r.t. $\widetilde{S}$ is defined as*

$$\widehat{\mathfrak{R}}_{\widetilde{S}}^*(L \circ \mathcal{F}) = \mathop{\mathbb{E}}_{\boldsymbol{\sigma}} \left[ \frac{1}{m} \sum_{j \in [J]} \omega_j \times \sup_{f \in \mathcal{F}} \left( \sum_{i \in I_j} \sigma_i L(\tilde{\boldsymbol{x}}_i^+, \tilde{\boldsymbol{x}}_i^-, f_{\alpha_i}, f_{\beta_i}) \right) \right],$$

*where $\boldsymbol{\sigma} = (\sigma_i)_{i=1}^n$, in which $\sigma_i$s are independent Rademacher variables.*

*For the hypothesis space $\mathcal{F}$, the empirical fractional Rademacher complexity of the hypothesis space w.r.t. the positive and negative parts of $\widetilde{S}$ is defined as follows, respectively:*

$$\widehat{\mathfrak{R}}_{\widetilde{S},+}^*(\mathcal{F}) = \mathop{\mathbb{E}}_{\boldsymbol{\sigma}} \left[ \frac{1}{m} \sum_{j \in [J]} \omega_j \left( \sup_{f \in \mathcal{F}} \sum_{i \in I_j} \sigma_i f_{\alpha_i}(\tilde{\boldsymbol{x}}_i^+) \right) \right],$$

$$\widehat{\mathfrak{R}}_{\widetilde{S},-}^*(\mathcal{F}) = \mathop{\mathbb{E}}_{\boldsymbol{\sigma}} \left[ \frac{1}{m} \sum_{j \in [J]} \omega_j \left( \sup_{f \in \mathcal{F}} \sum_{i \in I_j} \sigma_i f_{\beta_i}(\tilde{\boldsymbol{x}}_i^-) \right) \right].$$

Next, we give the base theorem of Micro-AUC for the subsequent generalization analyses.

**Theorem 1** (**The base theorem of Micro-AUC**). *Assume the loss function $L_\phi : \mathcal{X} \times \mathcal{X} \times \mathcal{F}_i \times \mathcal{F}_j \to \mathbb{R}_+$ is bounded by $M$. Then, for any $\delta > 0$, the following generalization bound holds with probability at least $1 - \delta$ over the draw of an i.i.d. sample $S$ of size $n$:*

$$\forall f \in \mathcal{F}, \ R_\phi(f) \leq \widehat{R}_S^\phi(f) + 2\widehat{\mathfrak{R}}_{\widetilde{S}}^*(L_\phi \circ \mathcal{F}) + 3M\sqrt{\frac{1}{2nK} \log\left(\frac{2}{\delta}\right)} \sqrt{\frac{1}{\tau_S}}.$$

Then, we analyze the relationship between $0/1$ and surrogate losses (see Appendix B.2.1). Further, we can get the relationship between $0/1$ and surrogate risks.

---

[2]In the following, we usually said it is balanced (or extremely imbalanced) for simplicity. Note that the multi-class dataset can be viewed as an extremely imbalanced one of MLL.

**Lemma 1** (**The relationship between** $0/1$ **and surrogate risks**)**.** *Assume the base loss $\ell$ upper bounds the original $0/1$ loss, i.e., $\mathbf{1}_{t\leq 0} \leq \ell(t)$. Then, $\forall S \overset{i.i.d.}{\sim} P$ and $f \in \mathcal{F}$, we can have*

$$R_{0/1}(f) \leq R_{pa}(f),$$

$$R_{0/1}(f) \leq R_{u_2}(f) = \underset{S}{\mathbb{E}}\left[\widehat{R}_S^{u_2}(f)\right] \leq \underset{S}{\mathbb{E}}\left[\frac{1}{\tau_S}\widehat{R}_S^{u_1}(f)\right] \leq \underset{S}{\mathbb{E}}\left[\frac{1-\tau_S}{\tau_S}\widehat{R}_S^{u_2}(f)\right].$$

**Remark.** *From the above lemma, we can see that when minimizing the surrogate risk, we can also minimize the $0/1$ risk. Additionally, the bound involving the $L_{u_1}$ and $L_{u_2}$ in the second inequality is tight since the equality holds when $\tau_S = \frac{1}{2}$.*

## 4.1 GENERAL DATASET

In this subsection, we give the generalization analyses of algorithms in the case of general datasets.

For the algorithm $\mathcal{A}^{pa}$, we can have a learning guarantee w.r.t. the Micro-AUC as follows.

**Theorem 2** (**Learning guarantee of $\mathcal{A}^{pa}$ for general datasets**)**.** *Suppose the surrogate loss $L_\phi = L_{pa}$ defined in Eq.(4) and Assumption 1 holds. Then, for any $\delta > 0$, the following generalization bound holds with probability at least $1 - \delta$:*

$$R_{0/1}(f) \leq R_{pa}(f) \leq \widehat{R}_{pa}(f) + 2\rho\left(\widehat{\mathfrak{R}}_{\widetilde{S},+}^*(\mathcal{F}) + \widehat{\mathfrak{R}}_{\widetilde{S},-}^*(\mathcal{F})\right) + 3B\sqrt{\frac{\log\left(\frac{2}{\delta}\right)}{2nK}}\sqrt{\frac{1}{\tau_S}}. \tag{13}$$

**Remark.** *For the second term involving the fractional Rademacher complexity, it can usually be upper bounded by the order of $O(\sqrt{\frac{1}{nK\tau_S}})$, e.g., the kernel-based one illustrated in Section 5.*

From this theorem, we can see that $\mathcal{A}^{pa}$ has a learning guarantee of $O(\sqrt{\frac{1}{nK\tau_S}})$ w.r.t. Micro-AUC.

Then, the algorithm $\mathcal{A}^{u_1}$ has a learning guarantee w.r.t. Micro-AUC as follows.

**Theorem 3** (**Learning guarantee of $\mathcal{A}^{u_1}$ for general datasets**)**.** *Suppose the surrogate loss $L_\phi = \frac{1}{\tau_S}L_{u_1}$ defined in Eq.(8) and Assumption 1 holds. Then, for any $\delta > 0$, the following generalization bound holds with probability at least $1 - \delta$:*

$$R_{0/1}(f) \leq \frac{1}{\tau_S}\widehat{R}_{u_1}(f) + \frac{2\rho}{\tau_S}\underbrace{\left(\frac{|\mathbb{S}^+|}{nK}\widehat{\mathfrak{R}}_{\widetilde{S},+}^*(\mathcal{F}) + \frac{|\mathbb{S}^-|}{nK}\widehat{\mathfrak{R}}_{\widetilde{S},-}^*(\mathcal{F})\right)}_{\approx \frac{1}{2}\left(\widehat{\mathfrak{R}}_{\widetilde{S},+}^*(\mathcal{F}) + \widehat{\mathfrak{R}}_{\widetilde{S},-}^*(\mathcal{F})\right)} + \frac{3B}{\tau_S}\sqrt{\frac{\log\left(\frac{2}{\delta}\right)}{2nK}}\sqrt{\frac{1}{\tau_S}}. \tag{14}$$

From this theorem, we can see $\mathcal{A}^{u_1}$ has a learning guarantee of $O(\frac{1}{\tau_S}\sqrt{\frac{1}{nK\tau_S}})$ w.r.t. Micro-AUC.

Finally, the algorithm $\mathcal{A}^{u_2}$ has a learning guarantee w.r.t. Micro-AUC as follows.

**Theorem 4** (**Learning guarantee of $\mathcal{A}^{u_2}$ for general datasets**)**.** *Suppose the surrogate loss $L_\phi = L_{u_2}$ defined in Eq.(9) and Assumption 1 holds. Then, for any $\delta > 0$, the following generalization bound holds with probability at least $1 - \delta$:*

$$R_{0/1}(f) \leq R_{u_2}(f) \leq \widehat{R}_{u_2}(f) + 2\rho\left(\widehat{\mathfrak{R}}_{\widetilde{S},+}^*(\mathcal{F}) + \widehat{\mathfrak{R}}_{\widetilde{S},-}^*(\mathcal{F})\right) + 6B\sqrt{\frac{\log\left(\frac{2}{\delta}\right)}{2nK}}\sqrt{\frac{1}{\tau_S}}. \tag{15}$$

From this theorem, we can see that $\mathcal{A}^{u_2}$ has a learning guarantee of $O(\sqrt{\frac{1}{nK\tau_S}})$ w.r.t. Micro-AUC.

## 4.2 BALANCED DATASET

Here we consider the case of balanced datasets. Notably, in this case, the algorithms $\mathcal{A}^{u_1}$ and $\mathcal{A}^{u_2}$ are exactly the same, which should have the same generalization guarantees w.r.t. Micro-AUC, and it is corroborated by the following corollary.

**Corollary 1** (**Learning guarantee of $\mathcal{A}^{u_1}$ and $\mathcal{A}^{u_2}$ for balanced datasets**)**.** *Suppose Assumption 1 holds and the dataset $S$ is balanced. Besides, assume the surrogate loss $L_\phi = L_{u_2} = 2L_{u_1}$. Then,*

*the following generalization bound holds with probability at least $1 - \delta$:*

$$R_{0/1}(f) \leq R_{u_2}(f) = 2R_{u_1}(f) \leq \widehat{R}_{u_2}(f) + 2\rho\left(\widehat{\mathfrak{R}}^*_{\widetilde{S},+}(\mathcal{F}) + \widehat{\mathfrak{R}}^*_{\widetilde{S},-}(\mathcal{F})\right) + 6B\sqrt{\frac{\log\left(\frac{2}{\delta}\right)}{nK}}, \quad (16)$$

*where* $\widehat{R}_{u_2}(f) = 2\widehat{R}_{u_1}(f)$.

From this corollary, we can see that $\mathcal{A}^{u_1}$ and $\mathcal{A}^{u_2}$ have learning guarantees of $O(\sqrt{\frac{1}{nK}})$ w.r.t. Micro-AUC. Note that the same learning guarantees confirm the validity of our analyses.

### 4.3 COMPARISON AND DISCUSSION

Theoretically, a tighter generalization error bound usually suggests better performance in practice (Shalev-Shwartz & Ben-David, 2014; Mohri et al., 2018).[3] Here we analyze these learning algorithms under the same framework and the bounds between the true and surrogate risk (or loss) are tight. Hence, we can safely assess the performance of algorithms by comparing their generalization upper bounds. Specifically, we compare them in the following.

$\mathcal{A}^{pa}$ **vs** $\mathcal{A}^{u_1}$**.** $\mathcal{A}^{pa}$ usually has a tighter bound than $\mathcal{A}^{u_1}$. Given the same hypothesis space, $\widehat{R}_{pa}(f)$ is usually easier to train than $\widehat{R}_{u_1}(f)$, which results in that $\widehat{R}_{pa} \leq \frac{1}{\tau_S}\widehat{R}_{u_1}$.[4] For the model complexity term (i.e., the last two terms in these bounds), we can see that $\mathcal{A}^{pa}$ has an error bound of $O(\sqrt{\frac{1}{nK\tau_S}})$ while $\mathcal{A}^{u_1}$ depends on $O(\frac{1}{\tau_S}\sqrt{\frac{1}{nK\tau_S}})$.

$\mathcal{A}^{u_2}$ **vs** $\mathcal{A}^{u_1}$**.** Similarly, $\mathcal{A}^{u_2}$ usually has a tighter bound than $\mathcal{A}^{u_1}$. For the empirical term, $\widehat{R}_{u_2}$ is usually smaller than $\frac{1}{\tau_S}\widehat{R}_{u_1}$. For the model complexity term, we can see that $\mathcal{A}^{u_2}$ has an error bound of $O(\sqrt{\frac{1}{nK\tau_S}})$ while $\mathcal{A}^{u_1}$ depends on $O(\frac{1}{\tau_S}\sqrt{\frac{1}{nK\tau_S}})$.

$\mathcal{A}^{pa}$ **vs** $\mathcal{A}^{u_2}$**.** $\mathcal{A}^{pa}$ usually has a comparable bound to $\mathcal{A}^{u_2}$. For the empirical term, $\widehat{R}_{pa}$ is usually comparable to $\widehat{R}_{u_2}$. For the model complexity term, they depend on the same order of $O(\sqrt{\frac{1}{nK\tau_S}})$.

Overall, $\mathcal{A}^{pa}$ and $\mathcal{A}^{u_2}$ usually have tigher bounds than $\mathcal{A}^{u_1}$, which suggests they would probably perform better than $\mathcal{A}^{u_1}$, especially when the dataset is highly imbalanced (i.e., $\tau_S$ is very small). Besides, the comparable bounds of $\mathcal{A}^{pa}$ and $\mathcal{A}^{u_2}$ imply that they would probably perform comparably to each other. Experimental results also confirm our theory analyses.

**Consistency.** Besides the generalization, it is also important to characterize the consistency properties of these surrogate losses, which is very challenging to our knowledge and left as future work.

## 5 KERNEL MODEL

For simplicity, here we consider the kernel-based hypothesis space as an illustration of $\mathcal{F}$, which has been widely studied both in practice (Elisseeff & Weston, 2001; Boutell et al., 2004; Hariharan et al., 2010; Tan et al., 2020; Wu et al., 2020) and in theory (Wu & Zhu, 2020; Wu et al., 2021; 2023) in MLL. Let $\Phi : \mathcal{X} \to \mathbb{H}$ be a feature mapping associated with the kernel function $\kappa$, where $\kappa : \mathcal{X} \times \mathcal{X} \to \mathbb{R}$ is a Positive Definite Symmetric (PDS) kernel and $\mathbb{H}$ denotes its induced reproducing kernel Hilbert space (RKHS). Formally, the considered kernel-based hypothesis space is defined as

$$\mathcal{F}^{kernel} = \left\{\boldsymbol{x} \mapsto \boldsymbol{W}^\top \Phi(\boldsymbol{x}) : \boldsymbol{W} = [\boldsymbol{w}_1, \ldots, \boldsymbol{w}_K], \|\boldsymbol{w}_k\|_{\mathbb{H}} \leq \Lambda, \forall k \in [K]\right\}. \quad (17)$$

Based on the preceding theoretical results in Section 4.1, we can have the following corollary.

**Corollary 2** (**Learning guarantee of** $\mathcal{A}^{pa}$ **for the kernel-based hypothesis space**)**.** *Suppose the surrogate loss* $L_\phi = L_{pa}$ *defined in Eq.(4) and Assumption 1 holds. Besides, assume the hypothesis space* $\mathcal{F} = \mathcal{F}^{kernel}$ *defined in Eq.(17), and* $\forall \boldsymbol{x} \in \mathcal{X}, \exists r > 0, \kappa(\boldsymbol{x}, \boldsymbol{x}) \leq r^2$. *Then, for any* $\delta > 0$, *the*

---

[3]Notably when comparing bounds, it is usually more rational to compare the order of dependent factors rather than the absolute values.

[4]Although we cannot formally prove this, experimental results support it.

*following generalization bound holds with probability at least $1 - \delta$:*

$$R_{0/1}(f) \leq R_{pa}(f) \leq \widehat{R}_{pa}(f) + \frac{4\rho r \Lambda}{\sqrt{nK}} \sqrt{\frac{1}{\tau_S}} + 3B \sqrt{\frac{\log\left(\frac{2}{\delta}\right)}{2nK}} \sqrt{\frac{1}{\tau_S}}. \tag{18}$$

From this corollary, we can see that $\mathcal{A}^{pa}$ has an error bound of $O(\sqrt{\frac{1}{nK\tau_S}})$ w.r.t. Micro-AUC. Similarly, we can also get the kernel-based counterparts of $\mathcal{A}^{u_1}$ and $\mathcal{A}^{u_2}$ (see Appendix B.5). Note that similar analyses can be extended to the case of the deep neural network-based hypothesis space, based on the related research on the complexity of various hypothesis spaces (Anthony & Bartlett, 1999; Bartlett et al., 2017; Long & Sedghi, 2019; Ma et al., 2020), left as future work.

## 6  RELATED WORK

**Consistency.** Gao & Zhou (2013) investigated the consistency of various surrogates on the measures of Hamming and (partial) ranking loss in MLL. Further, Dembczynski et al. (2012) gave an explicit regret bound of a univariate surrogate on the partial ranking loss measure. For the F-measure in binary classification with application in the Macro-F measure in MLL, Ye et al. (2012) justified and connected two approaches: the empirical utility maximization (EUM) and the decision-theoretic approach (DTA), where recent work (Dembczyński et al., 2017) gave more descriptive names Population Utility (PU) and Expected Test Utility (ETU).[5] Further, Waegeman et al. (2014); Zhang et al. (2020) investigated the consistency of the F-measure with different estimations of the conditional distribution $P(\boldsymbol{y}|\boldsymbol{x})$ under the DTA. For precision@$k$ and recall@$k$ measures, Menon et al. (2019) studied the multi-label consistency of various reduction methods.

**Generalization.** For the measures of subset accuracy and Hamming loss, Wu & Zhu (2020) studied the generalization of various algorithms, and identified the *label number* playing a critical role in these bounds, which could explain the phenomena better than previous results. For the ranking loss measure, Wu et al. (2021) revisited the consistency and generalization, and identified the *instance-wise class imbalance* affecting the generalization bounds, which could explain the phenomena better than consistency. For the Macro-AUC measure, Wu et al. (2023) identified the *label-wise class imbalance* playing a vital role in generalization bounds.

## 7  EXPERIMENTS

For experiments, the primary goal is to corroborate our theory findings rather than illustrate the superior performance of our proposed algorithms. Since our theoretical results can cover various model forms, we mainly conduct experiments with linear and neural network-based models on tabular and image datasets, respectively. For a fair comparison, we adopt the regularized ERM algorithms with $l_2$ norm and the same logistic base loss. See Appendix C for detailed setup and additional results.

### 7.1  TABULAR DATA

The tabular benchmarks and empirical results are summarized in Table 2 and 3, respectively. Overall, from Table 3, we can see $\mathcal{A}^{pa}$ and $\mathcal{A}^{u_2}$ perform better than $\mathcal{A}^{u_1}$, which corroborates our theory results that $\mathcal{A}^{pa}$ and $\mathcal{A}^{u_2}$ have better learning guarantees than $\mathcal{A}^{u_1}$. Besides, $\mathcal{A}^{pa}$ performs comparably to $\mathcal{A}^{u_2}$, which also confirms the theory results that they have comparable learning guarantees.

Further, from Table 2 and 3, we can carefully analyze the effect of the matrix-wise class imbalance level on the performance. Specifically, for the first five datasets, the imbalance level (i.e., $\frac{1}{\tau_S}$) is small, and the performance gap among these three algorithms is also small. In contrast, for the last five datasets, the imbalance level is high, and the performance gap between $\mathcal{A}^{u_1}$ and $\mathcal{A}^{u_2}$ is big. This also corroborates our theory results that $\mathcal{A}^{pa}$ and $\mathcal{A}^{u_1}$ have error bounds of $O(\sqrt{\frac{1}{nK\tau_S}})$ while $\mathcal{A}^{u_1}$ depends on $O(\frac{1}{\tau_S}\sqrt{\frac{1}{nK\tau_S}})$.

### 7.2  IMAGE DATA

The (raw) image benchmark datasets and empirical results are summarized in Table 4 and 5, respectively. From Table 4 and 5, we can observe that the imbalance levels of these datasets are large, and

---

[5]Note that our generalization analyses are under the EUM (or PU) framework.

Table 2: Basic statistics of the tabular benchmark datasets. Denote the matrix-wise class imbalance-related quantities $\text{Imb}_1 = \sqrt{\frac{1}{\tau_S}}$, $\text{Imb}_2 = \frac{1}{\tau_S}$ and $\text{Imb}_3 = \frac{1}{\tau_S}\sqrt{\frac{1}{\tau_S}}$, respectively.

| Dataset | #Instance | #Feature | #Label | Domain | $\text{Imb}_1$ | $\text{Imb}_2$ | $\text{Imb}_3$ |
|---|---|---|---|---|---|---|---|
| CAL500 | 502 | 68 | 174 | music | 2.5 | 6.7 | 17.3 |
| emotions | 593 | 72 | 6 | music | 1.8 | 3.2 | 5.8 |
| image | 2000 | 294 | 5 | images | 2.0 | 4.0 | 8.1 |
| scene | 2407 | 294 | 6 | images | 2.4 | 5.6 | 13.2 |
| yeast | 2417 | 103 | 14 | biology | 1.8 | 3.3 | 6.0 |
| enron | 1702 | 1001 | 53 | text | 4.0 | 15.7 | 62.1 |
| rcv1-s1 | 6000 | 944 | 101 | text | 5.9 | 35.1 | 207.7 |
| bibtex | 7395 | 1836 | 159 | text | 8.1 | 66.2 | 538.6 |
| corel5k | 5000 | 499 | 374 | images | 10.3 | 106.2 | 1094.3 |
| delicious | 16105 | 500 | 983 | text(web) | 7.2 | 51.7 | 371.5 |

Table 3: Micro-AUC (mean $\pm$ std, the symbol . means $0$.) of all three linear model-based algorithms based on tabular benchmark datasets. The top two algorithms on each dataset are highlighted in bold and the top one is labeled with $\dagger$. Besides, "-" means that $\mathcal{A}^{pa}$ takes more than one week by using a GPU server on the corresponding datasets.

| Dataset | $\mathcal{A}^{pa}$ | $\mathcal{A}^{u_1}$ | $\mathcal{A}^{u_2}$ |
|---|---|---|---|
| CAL500 | $\mathbf{.7841 \pm .0049}^{\dagger}$ | $.7765 \pm .0065$ | $\mathbf{.7831 \pm .0057}$ |
| emotions | $\mathbf{.8555 \pm .0153}$ | $.8550 \pm .0146$ | $\mathbf{.8560 \pm .0167}^{\dagger}$ |
| image | $\mathbf{.8426 \pm .0069}$ | $.8370 \pm .0094$ | $\mathbf{.8443 \pm .0070}^{\dagger}$ |
| scene | $\mathbf{.9352 \pm .0033}^{\dagger}$ | $.9311 \pm .0039$ | $\mathbf{.9318 \pm .0023}$ |
| yeast | $\mathbf{.8301 \pm .0008}$ | $.8284 \pm .0016$ | $\mathbf{.8309 \pm .0007}^{\dagger}$ |
| enron | $\mathbf{.8697 \pm .0168}$ | $.8499 \pm .0128$ | $\mathbf{.8736 \pm .0015}^{\dagger}$ |
| rcv1-s1 | - | $.9211 \pm .0055$ | $\mathbf{.9526 \pm .0008}^{\dagger}$ |
| bibtex | - | $.8919 \pm .0053$ | $\mathbf{.9417 \pm .0001}^{\dagger}$ |
| corel5k | - | $.7684 \pm .0032$ | $\mathbf{.8146 \pm .0043}^{\dagger}$ |
| delicious | - | $.8680 \pm .0007$ | $\mathbf{.9028 \pm .0007}^{\dagger}$ |

Table 4: Statistics of the (raw) image benchmark datasets. The meanings of the matrix-wise class imbalance-related quantities are the same as the ones in Table 2. The shape of inputs is resized.

| Dataset | #Instance | #Pixel | #Label | $\text{Imb}_1$ | $\text{Imb}_2$ | $\text{Imb}_3$ |
|---|---|---|---|---|---|---|
| PASCAL VOC 2012 | 11540 | $300 \times 300$ | 20 | 3.6 | 13.0 | 47.0 |
| MSCOCO 2017 | 123287 | $300 \times 300$ | 80 | 5.2 | 27.5 | 144.2 |
| NUS-WIDE | 269648 | $300 \times 300$ | 81 | 6.6 | 43.3 | 285.4 |

Table 5: Micro-AUC of all three algorithms with the ResNet-34 model on image benchmark datasets.

| Dataset | $\mathcal{A}^{pa}$ | $\mathcal{A}^{u_1}$ | $\mathcal{A}^{u_2}$ |
|---|---|---|---|
| PASCAL VOC 2012 | $\mathbf{.8357}$ | $.7449$ | $\mathbf{.8448}^{\dagger}$ |
| MSCOCO 2017 | $\mathbf{.8715}$ | $.8165$ | $\mathbf{.9062}^{\dagger}$ |
| NUS-WIDE | $\mathbf{.9206}$ | $.8513$ | $\mathbf{.9329}^{\dagger}$ |

$\mathcal{A}^{pa}$ and $\mathcal{A}^{u_2}$ perform better than $\mathcal{A}^{u_1}$, which corroborates our theory results that $\mathcal{A}^{pa}$ and $\mathcal{A}^{u_2}$ have better learning guarantees than $\mathcal{A}^{u_1}$.

## 8 CONCLUSION AND DISCUSSION

Toward theoretically understanding the Micro-AUC maximization problem in MLL, this paper takes an initial step to characterize the learning guarantees of three surrogate loss-based algorithms. Theoretically, a critical data-dependent quantity, i.e., the *matrix-wise class imbalance*, is identified. The results of the imbalance-aware error bounds suggest that algorithms with the proposed pairwise and reweighted univariate loss can have better learning guarantees than the one with the commonly-used univariate loss, which probably indicates their superior performance in practice. Experimental results also confirm the theory findings. As a by-product, the proposed reweighted surrogate loss can be a good candidate in applications for its good theory guarantee along with efficiency.

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

CONTENTS OF APPENDIX

## A    TECHNICAL PRELIMINARIES

Our subsequent analyses need techniques of the graph theory on the fractional independent vertex cover, and the dependent graph associated with a sample. For these technical backgrounds, we refer the readers to a recent survey (Zhang & Amini, 2022) on this topic and Appendix A in recent work (Wu et al., 2023).

In this work, our analyses are mainly based on the following theorem which can handle the learning task with graph-dependent examples.

**Theorem A.1** ((Usunier et al., 2005; Amini & Usunier, 2015)). *Give a sample* $\widetilde{S} = \{(\tilde{\boldsymbol{x}}_i, \tilde{\boldsymbol{y}}_i)\}_{i=1}^m$, *where* $\tilde{\boldsymbol{x}}_i \in \widetilde{\mathcal{X}}$, $\tilde{\boldsymbol{y}}_i \in \widetilde{\mathcal{Y}}$ *and* $\widetilde{S}$ *is associated with a dependency graph* $G$, *where* $\chi_f(G)$ *is its fractional chromatic number, and a loss function* $L : \widetilde{\mathcal{X}} \times \widetilde{\mathcal{Y}} \times \widetilde{\mathcal{F}} \to [0, M]$, *where* $\widetilde{\mathcal{F}} = \left\{ \tilde{f} : \widetilde{\mathcal{X}} \to \mathbb{R} \right\}$. *Then, for any* $\delta \in (0, 1)$, *the following generalization bound holds with probability at least* $1 - \delta$:

$$\forall f \in \mathcal{F}, \ R(f) \le \widehat{R}_{\widetilde{S}}(\tilde{f}) + 2\widehat{\mathfrak{R}}_{\widetilde{S}}^*(L \circ \widetilde{\mathcal{F}}) + 3M\sqrt{\frac{\chi_f(G)}{2m}\log\left(\frac{2}{\delta}\right)}, \qquad (19)$$

*where* $\widehat{\mathfrak{R}}_{\widetilde{S}}^*(L \circ \widetilde{\mathcal{F}})$ *is the empirical fractional Rademacher complexity of the loss space.*

## B    MAIN PROOFS

### B.1    PROOF OF THEOREM 1

#### B.1.1    PROBLEM TRANSFORMATION

For the Micro-AUC maximization problem in multi-label learning, we can transform it into the following learning task with graph-dependent examples.

Specifically, given a training dataset $S = ((\boldsymbol{x}_1, \boldsymbol{y}_1), \ldots, (\boldsymbol{x}_n, \boldsymbol{y}_n))$, which is an i.i.d. sample drawn from a MLL distribution $P$, where $\boldsymbol{x} \in \mathcal{X}$, and $\boldsymbol{y} \in \{-1, +1\}^K$, construct a dataset $\widetilde{S} = \{(\tilde{\boldsymbol{x}}_b, 1)\}_{b=1}^m = \{((\tilde{\boldsymbol{x}}_b^+, \tilde{\boldsymbol{x}}_b^-, \alpha_b, \beta_b), 1)\}_{b=1}^m$, where $\tilde{\boldsymbol{x}}_b^+ = \boldsymbol{x}_p$, $\tilde{\boldsymbol{x}}_b^- = \boldsymbol{x}_q$ for any $p, q \in [n]$, and $\alpha_b, \beta_b \in [K]$, $y_{p\alpha_b} = 1$, and $y_{q\beta_b} = -1$. Then, we can get that $m = |\mathbb{S}^+||\mathbb{S}^-|$. Denote $\widetilde{\mathcal{X}} = \mathcal{X} \times \mathcal{X} \times [K] \times [K]$ and given a loss function $L : \widetilde{\mathcal{X}} \times \{1\} \times \widetilde{\mathcal{F}} \to \mathbb{R}_+$, where the adjusted hypothesis space $\widetilde{\mathcal{F}} = \left\{ \tilde{f}(\tilde{\boldsymbol{x}}_i) = f_{\alpha_i}(\tilde{\boldsymbol{x}}_i^+) - f_{\beta_i}(\tilde{\boldsymbol{x}}_i^-) \mid f = (f_1, \ldots, f_K) : \mathcal{X} \to \mathbb{R}^K \in \mathcal{F} \right\}$. The goal of this new learning task is to learn a good mapping function $\tilde{f} : \widetilde{\mathcal{X}} \to \mathbb{R}$ from $\widetilde{\mathcal{F}}$ w.r.t. the loss function $L$.

Let $\{(I_j, \omega_j)\}_{j \in [J]}$ be a fractional independent vertex cover of the dependence graph $G$ constructed over $\widetilde{S}$ with $\sum_{j \in [J]} \omega_j = \chi_f(G)$, where $\chi_f(G)$ is the fractional chromatic number of $G$. Based on the previous results in bipartite ranking (Usunier et al., 2005; Amini & Usunier, 2015), we can know that

$$\chi_f(G) = \max\{|\mathbb{S}^+|, |\mathbb{S}^-|\} = (1 - \tau_S)nK.$$

#### B.1.2    PROOF OF THEOREM 1

*Proof.* Through the problem transformation in Section B.1.1, we can straightforwardly get the desired result by applying Theorem A.1. ☐

### B.2    PROOF OF LEMMA B.2.1 AND 1

#### B.2.1    PROOF OF LEMMA B.2.1

**Lemma B.2.1** (**The relationship between** $0/1$ **and surrogate losses**). *Assume the base loss* $\ell$ *upper bounds the original* $0/1$ *loss, i.e.,* $\mathbf{1}_{t \le 0} \le \ell(t)$. *Then,* $\forall (p, i) \in \mathbb{S}^+, (q, j) \in \mathbb{S}^-$, *and* $f \in \mathcal{F}$, *we can*

*have*

$$L_{0/1}(\boldsymbol{x}_p, \boldsymbol{x}_q, f_i, f_j) \leq L_{pa}(\boldsymbol{x}_p, \boldsymbol{x}_q, f_i, f_j),$$

$$L_{0/1}(\boldsymbol{x}_p, \boldsymbol{x}_q, f_i, f_j) \leq L_{u_2}(\boldsymbol{x}_p, \boldsymbol{x}_q, f_i, f_j) \leq \frac{1}{\tau_S} L_{u_1}(\boldsymbol{x}_p, \boldsymbol{x}_q, f_i, f_j) \leq \frac{1 - \tau_S}{\tau_S} L_{u_2}(\boldsymbol{x}_p, \boldsymbol{x}_q, f_i, f_j).$$

**Remark.** *From the above lemma, we can see that when minimizing the surrogate loss, we can also minimize the $0/1$ loss. Additionally, the bound involving the $L_{u_1}$ and $L_{u_2}$ in the second inequality is tight since the equality holds when $\tau_S = \frac{1}{2}$.*

*Proof.* For the first inequality, the following holds:

$$L_{0/1}(\boldsymbol{x}_p, \boldsymbol{x}_q, f_i, f_j) = \mathbf{1}_{f_i(\boldsymbol{x}_p) \leq f_j(\boldsymbol{x}_q)} \leq \ell(f_i(\boldsymbol{x}_p) - f_j(\boldsymbol{x}_q)) = L_{pa}(\boldsymbol{x}_p, \boldsymbol{x}_q, f_i, f_j).$$

For the second inequality, the following holds:

$$
\begin{aligned}
L_{0/1}(\boldsymbol{x}_p, \boldsymbol{x}_q, f_i, f_j) &= \mathbf{1}_{f_i(\boldsymbol{x}_p) \leq f_j(\boldsymbol{x}_q)} \\
&\leq \mathbf{1}_{\mathrm{sign}(f_i(\boldsymbol{x}_p)) \leq \mathrm{sign}(f_j(\boldsymbol{x}_q))} \\
&\overset{\textcircled{1}}{=} \mathbf{1}_{\mathrm{sign}(f_i(\boldsymbol{x}_p)) \neq +1} + \mathbf{1}_{\mathrm{sign}(f_j(\boldsymbol{x}_q)) \neq -1} - \mathbf{1}_{\mathrm{sign}(f_i(\boldsymbol{x}_p)) \neq +1} \mathbf{1}_{\mathrm{sign}(f_j(\boldsymbol{x}_q)) \neq -1} \\
&\leq \mathbf{1}_{\mathrm{sign}(f_i(\boldsymbol{x}_p)) \neq +1} + \mathbf{1}_{\mathrm{sign}(f_j(\boldsymbol{x}_q)) \neq -1} \\
&\leq \ell(f_i(\boldsymbol{x}_p)) + \ell(-f_j(\boldsymbol{x}_q)) \\
&= L_{u_2}(\boldsymbol{x}_p, \boldsymbol{x}_q, f_i, f_j) \\
&= \frac{nK}{\min\{|\mathbb{S}^+|, |\mathbb{S}^-|\}} \left( \frac{\min\{|\mathbb{S}^+|, |\mathbb{S}^-|\}}{nK} \ell(f_i(\boldsymbol{x}_p)) + \frac{\min\{|\mathbb{S}^+|, |\mathbb{S}^-|\}}{nK} \ell(-f_j(\boldsymbol{x}_q)) \right) \\
&\leq \frac{1}{\tau_S} \left( \frac{|\mathbb{S}^+|}{nK} \ell(f_i(\boldsymbol{x}_p)) + \frac{|\mathbb{S}^-|}{nK} \ell(-f_j(\boldsymbol{x}_q)) \right) \\
&= \frac{1}{\tau_S} L_{u_1}(\boldsymbol{x}_p, \boldsymbol{x}_q, f_i, f_j) \\
&\leq \frac{1}{\tau_S} \left( \frac{\max\{|\mathbb{S}^+|, |\mathbb{S}^-|\}}{nK} \ell(f_i(\boldsymbol{x}_p)) + \frac{\max\{|\mathbb{S}^+|, |\mathbb{S}^-|\}}{nK} \ell(-f_j(\boldsymbol{x}_q)) \right) \\
&= \frac{\max\{|\mathbb{S}^+|, |\mathbb{S}^-|\}}{\min\{|\mathbb{S}^+|, |\mathbb{S}^-|\}} \left( \ell(f_i(\boldsymbol{x}_p)) + \ell(-f_j(\boldsymbol{x}_q)) \right) \\
&= \frac{1 - \tau_S}{\tau_S} L_{u_2}(\boldsymbol{x}_p, \boldsymbol{x}_q, f_i, f_j).
\end{aligned}
$$

For $\textcircled{1}$, we can enumerate the possible values of $\mathrm{sign}(f_i(\boldsymbol{x}_p))$ and $\mathrm{sign}(f_j(\boldsymbol{x}_q))$ to get the equality. $\square$

### B.2.2 PROOF OF LEMMA 1

*Proof.* For the first inequality, the following holds:

$$
\begin{aligned}
R_{0/1}(f) = \mathbb{E}_S \left[ \widehat{R}_S^{0/1}(f) \right] &= \mathbb{E}_S \left[ \frac{1}{|\mathbb{S}^+||\mathbb{S}^-|} \sum_{i=1}^K \sum_{j=1}^K \sum_{(p,q) \in \mathbb{S}_i^+ \times \mathbb{S}_j^-} L_{0/1}(\boldsymbol{x}_p, \boldsymbol{x}_q, f_i, f_j) \right] \\
&\overset{\textcircled{1}}{\leq} \mathbb{E}_S \left[ \frac{1}{|\mathbb{S}^+||\mathbb{S}^-|} \sum_{i=1}^K \sum_{j=1}^K \sum_{(p,q) \in \mathbb{S}_i^+ \times \mathbb{S}_j^-} L_{pa}(\boldsymbol{x}_p, \boldsymbol{x}_q, f_i, f_j) \right] \\
&= \mathbb{E}_S \left[ \widehat{R}_S^{pa}(f) \right] \\
&= R_{pa}(f).
\end{aligned}
$$

For $\textcircled{1}$, it is due to the first inequality in Lemma B.2.1.

For the second inequality, the following holds:

$$R_{0/1}(f) = \mathop{\mathbb{E}}_{S}\left[\widehat{R}_S^{0/1}(f)\right] = \mathop{\mathbb{E}}_{S}\left[\frac{1}{|\mathbb{S}^+||\mathbb{S}^-|}\sum_{i=1}^{K}\sum_{j=1}^{K}\sum_{(p,q)\in\mathbb{S}_i^+\times\mathbb{S}_j^-}L_{0/1}(\boldsymbol{x}_p,\boldsymbol{x}_q,f_i,f_j)\right]$$

$$\overset{\textcircled{2}}{\leq}\mathop{\mathbb{E}}_{S}\left[\frac{1}{|\mathbb{S}^+||\mathbb{S}^-|}\sum_{i=1}^{K}\sum_{j=1}^{K}\sum_{(p,q)\in\mathbb{S}_i^+\times\mathbb{S}_j^-}L_{u_2}(\boldsymbol{x}_p,\boldsymbol{x}_q,f_i,f_j)\right]$$

$$=\mathop{\mathbb{E}}_{S}\left[\widehat{R}_S^{u_2}(f)\right]$$

$$=R_{u_2}(f)$$

$$\overset{\textcircled{3}}{\leq}\mathop{\mathbb{E}}_{S}\left[\frac{1}{\tau_S|\mathbb{S}^+||\mathbb{S}^-|}\sum_{i=1}^{K}\sum_{j=1}^{K}\sum_{(p,q)\in\mathbb{S}_i^+\times\mathbb{S}_j^-}L_{u_1}(\boldsymbol{x}_p,\boldsymbol{x}_q,f_i,f_j)\right]$$

$$=\mathop{\mathbb{E}}_{S}\left[\frac{1}{\tau_S}\widehat{R}_S^{u_1}(f)\right]$$

$$\overset{\textcircled{4}}{\leq}\mathop{\mathbb{E}}_{S}\left[\frac{1-\tau_S}{\tau_S|\mathbb{S}^+||\mathbb{S}^-|}\sum_{i=1}^{K}\sum_{j=1}^{K}\sum_{(p,q)\in\mathbb{S}_i^+\times\mathbb{S}_j^-}L_{u_2}(\boldsymbol{x}_p,\boldsymbol{x}_q,f_i,f_j)\right]$$

$$=\mathop{\mathbb{E}}_{S}\left[\frac{1-\tau_S}{\tau_S}\widehat{R}_S^{u_2}(f)\right].$$

For ②, ③ and ④, they are due to the second inequality in Lemma B.2.1. □

### B.3  THE LEMMAS OF THE CONTRACTION INEQUALITIES

**Lemma B.3.1 (The base contraction inequality).** *Assume the base loss $\ell(\cdot)$ is $\rho$-Lipschitz continuous. Then, the following inequalities hold:*

$$\mathop{\mathbb{E}}_{\boldsymbol{\sigma}}\left[\frac{1}{m}\sum_{j\in[J]}\omega_j\left(\sup_{f\in\mathcal{F}}\sum_{i\in I_j}\sigma_i\ell(f_{\alpha_i}(\tilde{\boldsymbol{x}}_i^+))\right)\right]\leq\rho\mathop{\mathbb{E}}_{\boldsymbol{\sigma}}\left[\frac{1}{m}\sum_{j\in[J]}\omega_j\left(\sup_{f\in\mathcal{F}}\sum_{i\in I_j}\sigma_i f_{\alpha_i}(\tilde{\boldsymbol{x}}_i^+)\right)\right],\quad(20)$$

$$\mathop{\mathbb{E}}_{\boldsymbol{\sigma}}\left[\frac{1}{m}\sum_{j\in[J]}\omega_j\left(\sup_{f\in\mathcal{F}}\sum_{i\in I_j}\sigma_i\ell(-f_{\beta_i}(\tilde{\boldsymbol{x}}_i^-))\right)\right]\leq\rho\mathop{\mathbb{E}}_{\boldsymbol{\sigma}}\left[\frac{1}{m}\sum_{j\in[J]}\omega_j\left(\sup_{f\in\mathcal{F}}\sum_{i\in I_j}\sigma_i f_{\beta_i}(\tilde{\boldsymbol{x}}_i^-)\right)\right],$$

$$(21)$$

$$\mathop{\mathbb{E}}_{\boldsymbol{\sigma}}\left[\frac{1}{m}\sum_{j\in[J]}\omega_j\left(\sup_{f\in\mathcal{F}}\sum_{i\in I_j}\sigma_i\ell(f_{\alpha_i}(\tilde{\boldsymbol{x}}_i^+)-f_{\beta_i}(\tilde{\boldsymbol{x}}_i^-))\right)\right]\leq$$

$$\rho\mathop{\mathbb{E}}_{\boldsymbol{\sigma}}\left[\frac{1}{m}\sum_{j\in[J]}\omega_j\left(\sup_{f\in\mathcal{F}}\sum_{i\in I_j}\sigma_i(f_{\alpha_i}(\tilde{\boldsymbol{x}}_i^+)-f_{\beta_i}(\tilde{\boldsymbol{x}}_i^-))\right)\right].\quad(22)$$

*Proof.* Here we mainly prove the first inequality following the idea in Mohri et al. (2018) (Lemma 5.7, p.93), and the other two inequalities share the same proof ideas, which are omitted for brevity.

First we fix a sample $(\tilde{\boldsymbol{x}}_1, \ldots, \tilde{\boldsymbol{x}}_m)$, then by definition,

$$\underset{\boldsymbol{\sigma}}{\mathbb{E}}\left[\frac{1}{m}\sum_{j\in[J]}\omega_j\sup_{f\in\mathcal{F}}\left(\sum_{i\in I_j}\sigma_i\ell(f_{\alpha_i}(\tilde{\boldsymbol{x}}_i^+))\right)\right]$$

$$=\frac{1}{m}\sum_{j\in[J]}\omega_j\underset{\sigma_1,\ldots,\sigma_{n_j-1}}{\mathbb{E}}\left[\underset{\sigma_{n_j}}{\mathbb{E}}\left[\sup_{f\in\mathcal{F}}u_{n_j-1}(f)+\sigma_{n_j}\ell(f(\tilde{\boldsymbol{x}}_{n_j}^+))\right]\right],\quad(\text{denote }n_j=|I_j|\text{ for simplicity})$$

where $u_{n_j-1}(f)=\sum_{i=1}^{n_j}\sigma_i\ell(f(\tilde{\boldsymbol{x}}_i^+))$. By the definition of the supremum, for any $\epsilon > 0$, there exists $f^1, f^2 \in \mathcal{F}$ such that

$$u_{n_j-1}(f^1)+\ell(f^1(\tilde{\boldsymbol{x}}_{n_j}^+))\le(1-\epsilon)\left[\sup_{f\in\mathcal{F}}u_{n_j-1}(f)+\ell(f(\tilde{\boldsymbol{x}}_{n_j}^+))\right],$$

and

$$u_{n_j-1}(f^2)-\ell(f^2(\tilde{\boldsymbol{x}}_{n_j}^+))\le(1-\epsilon)\left[\sup_{f\in\mathcal{F}}u_{n_j-1}(f)-\ell(f(\tilde{\boldsymbol{x}}_{n_j}^+))\right].$$

Thus, for any $\epsilon > 0$, by definition of $\underset{\sigma_{n_j}}{\mathbb{E}}$,

$$(1-\epsilon)\underset{\sigma_{n_j}}{\mathbb{E}}\left[\sup_{f\in\mathcal{F}}u_{n_j-1}(f)+\sigma_{n_j}\ell(f(\tilde{\boldsymbol{x}}_{n_j}^+))\right]$$

$$=(1-\epsilon)\left[\frac{1}{2}\sup_{f\in\mathcal{F}}u_{n_j-1}(f)+\ell(f(\tilde{\boldsymbol{x}}_{n_j}^+))\right]+\left[\frac{1}{2}\sup_{f\in\mathcal{F}}u_{n_j-1}(f)-\ell(f(\tilde{\boldsymbol{x}}_{n_j}^+))\right]$$

$$\le\frac{1}{2}\left[u_{n_j-1}(f^1)+\ell(f^1(\tilde{\boldsymbol{x}}_{n_j}^+))\right]+\frac{1}{2}\left[u_{n_j-1}(f^2)-\ell(f^2(\tilde{\boldsymbol{x}}_{n_j}^+))\right].$$

Let $s=\operatorname{sign}(f^1(\boldsymbol{x}_{n_j})-f^2(\boldsymbol{x}_{n_j}))$. Then, the previous inequality implies

$$(1-\epsilon)\underset{\sigma_{n_j}}{\mathbb{E}}\left[\sup_{f\in\mathcal{F}}u_{n_j-1}(f)+\sigma_{n_j}\ell(f(\tilde{\boldsymbol{x}}_{n_j}^+))\right]$$

$$\le\frac{1}{2}\left[u_{n_j-1}(f^1)+u_{n_j-1}(f^2)+s\rho(f^1(\tilde{\boldsymbol{x}}_{n_j}^+)-f^2(\tilde{\boldsymbol{x}}_{n_j}^+))\right]\qquad(\text{Lipschitz property})$$

$$=\frac{1}{2}\left[u_{n_j-1}(f^1)+s\rho f^1(\tilde{\boldsymbol{x}}_{n_j}^+)\right]+\frac{1}{2}\left[u_{n_j-1}(f^2)-s\rho f^2(\tilde{\boldsymbol{x}}_{n_j}^+)\right]\qquad(\text{rearranging})$$

$$\le\frac{1}{2}\sup_{f\in\mathcal{F}}\left[u_{n_j-1}(f)+s\rho f(\tilde{\boldsymbol{x}}_{n_j}^+)\right]+\frac{1}{2}\sup_{f\in\mathcal{F}}\left[u_{n_j-1}(f)-s\rho f(\tilde{\boldsymbol{x}}_{n_j}^+)\right]\qquad(\text{definition of sup})$$

$$=\underset{\sigma_{n_j}}{\mathbb{E}}\left[\sup_{f\in\mathcal{F}}u_{n_j-1}(f)+\sigma_{n_j}\mu f(\tilde{\boldsymbol{x}}_{n_j}^+)\right].\qquad(\text{definition of }\underset{\sigma_{n_j}}{\mathbb{E}})$$

Since the inequality holds for any $\epsilon > 0$, we have

$$\underset{\sigma_{n_j}}{\mathbb{E}}\left[\sup_{f\in\mathcal{F}}u_{n_j-1}(f)+\sigma_{n_j}\ell(f(\tilde{\boldsymbol{x}}_{n_j}^+))\right]\le\underset{\sigma_{n_j}}{\mathbb{E}}\left[\sup_{f\in\mathcal{F}}u_{n_j-1}(f)+\sigma_{n_j}\mu f(\tilde{\boldsymbol{x}}_{n_j}^+)\right].$$

Proceeding in the same way for all other $\sigma_i$ ($i\in[I_j], i\ne n_j$) proves that

$$\underset{\boldsymbol{\sigma}}{\mathbb{E}}\left[\sup_{f\in\mathcal{F}}\left(\sum_{i\in I_j}\sigma_i\ell(f(\tilde{\boldsymbol{x}}_i^+))\right)\right]\le\rho\underset{\boldsymbol{\sigma}}{\mathbb{E}}\left[\sup_{f\in\mathcal{F}}\left(\sum_{i\in I_j}\sigma_i f(\tilde{\boldsymbol{x}}_i^+)\right)\right].$$

By proceeding other $j \in [J]$, we can obtain the following

$$\frac{1}{m}\sum_{j\in[J]}\omega_j\underset{\boldsymbol{\sigma}}{\mathbb{E}}\left[\sup_{f\in\mathcal{F}}\left(\sum_{i\in I_j}\sigma_i\ell(f(\tilde{\boldsymbol{x}}_i^+))\right)\right]\le\rho\frac{1}{m}\sum_{j\in[J]}\omega_j\underset{\boldsymbol{\sigma}}{\mathbb{E}}\left[\sup_{f\in\mathcal{F}}\left(\sum_{i\in I_j}\sigma_i f(\tilde{\boldsymbol{x}}_i^+)\right)\right].$$

Then, by the linearity of expectation, we can get the first inequality. □

**Lemma B.3.2 (Contraction inequality for $L_{pa}$).** *Assume the surrogate loss $L_\phi = L_{pa}$ defined in Eq.(4) and the base loss $\ell(\cdot)$ is $\rho$-Lipschitz continuous. Then, the following inequality holds:*

$$\widehat{\mathfrak{R}}^*_{\widetilde{S}}(L_{pa} \circ \mathcal{F}) \leq \rho \left( \widehat{\mathfrak{R}}^*_{\widetilde{S},+}(\mathcal{F}) + \widehat{\mathfrak{R}}^*_{\widetilde{S},-}(\mathcal{F}) \right). \tag{23}$$

*Proof.* For this inequality, we can have that

$$\widehat{\mathfrak{R}}^*_{\widetilde{S}}(L_{pa} \circ \mathcal{F}) = \mathop{\mathbb{E}}_{\boldsymbol{\sigma}} \left[ \frac{1}{m} \sum_{j \in [J]} \omega_j \left( \sup_{f \in \mathcal{F}} \sum_{i \in I_j} \sigma_i \ell(f_{\alpha_i}(\tilde{\boldsymbol{x}}_i^+) - f_{\beta_i}(\tilde{\boldsymbol{x}}_i^-)) \right) \right]$$

$$\overset{\textcircled{1}}{\leq} \rho \mathop{\mathbb{E}}_{\boldsymbol{\sigma}} \left[ \frac{1}{m} \sum_{j \in [J]} \omega_j \left( \sup_{f \in \mathcal{F}} \sum_{i \in I_j} \sigma_i (f_{\alpha_i}(\tilde{\boldsymbol{x}}_i^+) - f_{\beta_i}(\tilde{\boldsymbol{x}}_i^-)) \right) \right]$$

$$\overset{\textcircled{2}}{\leq} \rho \left( \mathop{\mathbb{E}}_{\boldsymbol{\sigma}} \left[ \frac{1}{m} \sum_{j \in [J]} \omega_j \left( \sup_{f \in \mathcal{F}} \sum_{i \in I_j} \sigma_i f_{\alpha_i}(\tilde{\boldsymbol{x}}_i^+) \right) \right] + \mathop{\mathbb{E}}_{\boldsymbol{\sigma}} \left[ \frac{1}{m} \sum_{j \in [J]} \omega_j \left( \sup_{f \in \mathcal{F}} \sum_{i \in I_j} \sigma_i f_{\beta_i}(\tilde{\boldsymbol{x}}_i^-) \right) \right] \right)$$

$$= \rho \left( \widehat{\mathfrak{R}}^*_{\widetilde{S},+}(\mathcal{F}) + \widehat{\mathfrak{R}}^*_{\widetilde{S},-}(\mathcal{F}) \right).$$

For $\textcircled{1}$, it is due to the third inequality in Lemma B.3.1. For $\textcircled{2}$, it is due to that $\sup(a+b) \leq \sup(a) + \sup(b)$, the property of the Rademacher variables, and the linearity of expectation. $\qquad \square$

**Lemma B.3.3 (Contraction inequality for $L_{u_1}$).** *Assume the surrogate loss $L_\phi = L_{u_1}$ defined in Eq.(8) and the base loss $\ell(\cdot)$ is $\rho$-Lipschitz continuous. Then, the following inequality holds:*

$$\widehat{\mathfrak{R}}^*_{\widetilde{S}}(L_{u_1} \circ \mathcal{F}) \leq \rho \left( \frac{|\mathbb{S}^+|}{nK} \widehat{\mathfrak{R}}^*_{\widetilde{S},+}(\mathcal{F}) + \frac{|\mathbb{S}^-|}{nK} \widehat{\mathfrak{R}}^*_{\widetilde{S},-}(\mathcal{F}) \right). \tag{24}$$

*Proof.* For this inequality, we have that

$$\widehat{\mathfrak{R}}^*_{\widetilde{S}}(L_{u_2} \circ \mathcal{F}) = \mathop{\mathbb{E}}_{\boldsymbol{\sigma}} \left[ \frac{1}{m} \sum_{j \in [J]} \omega_j \left( \sup_{f \in \mathcal{F}} \sum_{i \in I_j} \sigma_i \left( \frac{|\mathbb{S}^+|}{nK} \ell(f_{\alpha_i}(\tilde{\boldsymbol{x}}_i^+)) + \frac{|\mathbb{S}^-|}{nK} \ell(-f_{\beta_i}(\tilde{\boldsymbol{x}}_i^-)) \right) \right) \right]$$

$$\overset{\textcircled{1}}{\leq} \frac{|\mathbb{S}^+|}{nK} \mathop{\mathbb{E}}_{\boldsymbol{\sigma}} \left[ \frac{1}{m} \sum_{j \in [J]} \omega_j \left( \sup_{f \in \mathcal{F}} \sum_{i \in I_j} \sigma_i \ell(f_{\alpha_i}(\tilde{\boldsymbol{x}}_i^+)) \right) \right] + \frac{|\mathbb{S}^-|}{nK} \mathop{\mathbb{E}}_{\boldsymbol{\sigma}} \left[ \frac{1}{m} \sum_{j \in [J]} \omega_j \left( \sup_{f \in \mathcal{F}} \sum_{i \in I_j} \sigma_i \ell(-f_{\beta_i}(\tilde{\boldsymbol{x}}_i^-)) \right) \right]$$

$$\overset{\textcircled{2}}{\leq} \rho \left( \frac{|\mathbb{S}^+|}{nK} \mathop{\mathbb{E}}_{\boldsymbol{\sigma}} \left[ \frac{1}{m} \sum_{j \in [J]} \omega_j \left( \sup_{f \in \mathcal{F}} \sum_{i \in I_j} \sigma_i f_{\alpha_i}(\tilde{\boldsymbol{x}}_i^+) \right) \right] + \frac{|\mathbb{S}^-|}{nK} \mathop{\mathbb{E}}_{\boldsymbol{\sigma}} \left[ \frac{1}{m} \sum_{j \in [J]} \omega_j \left( \sup_{f \in \mathcal{F}} \sum_{i \in I_j} \sigma_i f_{\beta_i}(\tilde{\boldsymbol{x}}_i^-) \right) \right] \right)$$

$$= \rho \left( \frac{|\mathbb{S}^+|}{nK} \widehat{\mathfrak{R}}^*_{\widetilde{S},+}(\mathcal{F}) + \frac{|\mathbb{S}^-|}{nK} \widehat{\mathfrak{R}}^*_{\widetilde{S},-}(\mathcal{F}) \right).$$

For $\textcircled{1}$, it is due to that $\sup(a+b) \leq \sup(a) + \sup(b)$, and the linearity of expectation. For $\textcircled{2}$, it is due to the first and second inequality in Lemma B.3.1. $\qquad \square$

**Lemma B.3.4 (Contraction inequality for $L_{u_2}$).** *Assume the surrogate loss $L_\phi = L_{u_2}$ defined in Eq.(9) and the base loss $\ell(\cdot)$ is $\rho$-Lipschitz continuous. Then, the following inequality holds:*

$$\widehat{\mathfrak{R}}^*_{\widetilde{S}}(L_{u_2} \circ \mathcal{F}) \leq \rho \left( \widehat{\mathfrak{R}}^*_{\widetilde{S},+}(\mathcal{F}) + \widehat{\mathfrak{R}}^*_{\widetilde{S},-}(\mathcal{F}) \right). \tag{25}$$

*Proof.* For this inequality, we have that

$$
\widehat{\mathfrak{R}}_{\widetilde{S}}^{*}(L_{u_2} \circ \mathcal{F}) = \mathop{\mathbb{E}}_{\boldsymbol{\sigma}} \left[ \frac{1}{m} \sum_{j \in [J]} \omega_j \left( \sup_{f \in \mathcal{F}} \sum_{i \in I_j} \sigma_i \left( \ell(f_{\alpha_i}(\tilde{\boldsymbol{x}}_i^+)) + \ell(-f_{\beta_i}(\tilde{\boldsymbol{x}}_i^-)) \right) \right) \right]
$$

$$
\overset{\textcircled{1}}{\leq} \mathop{\mathbb{E}}_{\boldsymbol{\sigma}} \left[ \frac{1}{m} \sum_{j \in [J]} \omega_j \left( \sup_{f \in \mathcal{F}} \sum_{i \in I_j} \sigma_i \ell(f_{\alpha_i}(\tilde{\boldsymbol{x}}_i^+)) \right) \right] + \mathop{\mathbb{E}}_{\boldsymbol{\sigma}} \left[ \frac{1}{m} \sum_{j \in [J]} \omega_j \left( \sup_{f \in \mathcal{F}} \sum_{i \in I_j} \sigma_i \ell(-f_{\beta_i}(\tilde{\boldsymbol{x}}_i^-)) \right) \right]
$$

$$
\overset{\textcircled{2}}{\leq} \rho \left( \mathop{\mathbb{E}}_{\boldsymbol{\sigma}} \left[ \frac{1}{m} \sum_{j \in [J]} \omega_j \left( \sup_{f \in \mathcal{F}} \sum_{i \in I_j} \sigma_i f_{\alpha_i}(\tilde{\boldsymbol{x}}_i^+) \right) \right] + \mathop{\mathbb{E}}_{\boldsymbol{\sigma}} \left[ \frac{1}{m} \sum_{j \in [J]} \omega_j \left( \sup_{f \in \mathcal{F}} \sum_{i \in I_j} \sigma_i f_{\beta_i}(\tilde{\boldsymbol{x}}_i^-) \right) \right] \right)
$$

$$
= \rho \left( \widehat{\mathfrak{R}}_{\widetilde{S},+}^{*}(\mathcal{F}) + \widehat{\mathfrak{R}}_{\widetilde{S},-}^{*}(\mathcal{F}) \right).
$$

For ①, it is due to that $\sup(a + b) \leq \sup(a) + \sup(b)$, and the linearity of expectation. For ②, it is due to the first and second inequality in Lemma B.3.1. □

## B.4 PROOF OF THEOREM 2, 3 AND 4

### B.4.1 PROOF OF THEOREM 2

*Proof.* Since the surrogate loss $L_\phi = L_{pa}$ and Assumption 1 holds, we can get that $L_\phi$ is bounded by $B$. Then, applying Theorem 1 and Lemma B.3.2, we can get that, for any $\delta > 0$, the following generalization bound holds with probability at least $1 - \delta$:

$$
R_{pa}(f) \leq \widehat{R}_{pa}(f) + 2\rho \left( \widehat{\mathfrak{R}}_{\widetilde{S},+}^{*}(\mathcal{F}) + \widehat{\mathfrak{R}}_{\widetilde{S},-}^{*}(\mathcal{F}) \right) + 3B \sqrt{\frac{\log\left(\frac{2}{\delta}\right)}{2nK}} \sqrt{\frac{1}{\tau_S}}.
$$

Based on Lemma 1 (i.e., $R_{0/1}(f) \leq R_{pa}(f)$), we can get the desired result. □

### B.4.2 PROOF OF THEOREM 3

*Proof.* Since the surrogate loss $L_\phi = \frac{1}{\tau_S} L_{u_1}$ and Assumption 1 holds, we can get that $L_\phi$ is bounded by $\frac{B}{\tau_S}$. Then, applying Theorem 1 and Lemma B.3.3, we can get that, for any $\delta > 0$, the following generalization bound holds with probability at least $1 - \delta$:

$$
\mathop{\mathbb{E}}_{S} \left[ \frac{1}{\tau_S} \widehat{R}_S^{u_1}(f) \right] \leq \frac{1}{\tau_S} \widehat{R}_{u_1}(f) + \frac{2\rho}{\tau_S} \underbrace{\left( \frac{|\mathbb{S}^+|}{nK} \widehat{\mathfrak{R}}_{\widetilde{S},+}^{*}(\mathcal{F}) + \frac{|\mathbb{S}^-|}{nK} \widehat{\mathfrak{R}}_{\widetilde{S},-}^{*}(\mathcal{F}) \right)}_{\approx \frac{1}{2}\left( \widehat{\mathfrak{R}}_{\widetilde{S},+}^{*}(\mathcal{F}) + \widehat{\mathfrak{R}}_{\widetilde{S},-}^{*}(\mathcal{F}) \right)} + \frac{3B}{\tau_S} \sqrt{\frac{\log\left(\frac{2}{\delta}\right)}{2nK}} \sqrt{\frac{1}{\tau_S}}.
$$

Based on Lemma 1 (i.e., $R_{0/1}(f) \leq \mathop{\mathbb{E}}_S \left[ \frac{1}{\tau_S} \widehat{R}_S^{u_1}(f) \right]$), we can get the desired result. Besides, when $\tau_S = \frac{1}{2}$, the equality holds for the second approximation term involving the Rademacher complexity. □

### B.4.3 PROOF OF THEOREM 4

*Proof.* Since the surrogate loss $L_\phi = L_{u_2}$ and Assumption 1 holds, we can get that $L_\phi$ is bounded by $2B$. Then, applying Theorem 1 and Lemma B.3.4, we can get that, for any $\delta > 0$, the following generalization bound holds with probability at least $1 - \delta$:

$$
R_{u_2}(f) \leq \widehat{R}_{u_2}(f) + 2\rho \left( \widehat{\mathfrak{R}}_{\widetilde{S},+}^{*}(\mathcal{F}) + \widehat{\mathfrak{R}}_{\widetilde{S},-}^{*}(\mathcal{F}) \right) + 6B \sqrt{\frac{\log\left(\frac{2}{\delta}\right)}{2nK}} \sqrt{\frac{1}{\tau_S}}.
$$

Based on Lemma 1 (i.e., $R_{0/1}(f) \leq R_{u_2}(f)$), we can get the desired result. □

## B.5 Proof of Corollary 2, 3 and 4

**Corollary 3 (Learning guarantee of $\mathcal{A}^{u_1}$ for the kernel-based hypothesis space).** *Suppose the surrogate loss $L_\phi = \frac{1}{\tau_S} L_{u_1}$ defined in Eq.(8) and Assumption 1 holds. Besides, assume the hypothesis space $\mathcal{F} = \mathcal{F}^{kernel}$ defined in Eq.(17), and $\forall \boldsymbol{x} \in \mathcal{X}, \exists r > 0, \kappa(\boldsymbol{x}, \boldsymbol{x}) \leq r^2$. Then, for any $\delta > 0$, the following generalization bound holds with probability at least $1 - \delta$:*

$$R_{0/1}(f) \leq \frac{1}{\tau_S} \widehat{R}_{u_1}(f) + \frac{2\rho r \Lambda}{\tau_S \sqrt{nK}} \sqrt{\frac{1}{\tau_S}} + \frac{3B}{\tau_S} \sqrt{\frac{\log\left(\frac{2}{\delta}\right)}{2nK}} \sqrt{\frac{1}{\tau_S}}. \tag{26}$$

**Corollary 4 (Learning guarantee of $\mathcal{A}^{u_2}$ for the kernel-based hypothesis space).** *Suppose the surrogate loss $L_\phi = L_{u_2}$ defined in Eq.(9) and Assumption 1 holds. Besides, assume the hypothesis space $\mathcal{F} = \mathcal{F}^{kernel}$ defined in Eq.(17), and $\forall \boldsymbol{x} \in \mathcal{X}, \exists r > 0, \kappa(\boldsymbol{x}, \boldsymbol{x}) \leq r^2$. Then, for any $\delta > 0$, the following generalization bound holds with probability at least $1 - \delta$:*

$$R_{0/1}(f) \leq R_{u_2}(f) \leq \widehat{R}_{u_2}(f) + \frac{4\rho r \Lambda}{\sqrt{nK}} \sqrt{\frac{1}{\tau_S}} + 6B \sqrt{\frac{\log\left(\frac{2}{\delta}\right)}{2nK}} \sqrt{\frac{1}{\tau_S}}. \tag{27}$$

**Lemma B.5.1 (The Rademacher complexity of the kernel-based hypothesis space).** *Assume the hypothesis space $\mathcal{F} = \mathcal{F}^{kernel}$ defined in Eq.(17), and $\forall \boldsymbol{x} \in \mathcal{X}, \exists r > 0, \kappa(\boldsymbol{x}, \boldsymbol{x}) \leq r^2$. Then, we have*

$$\widehat{\mathfrak{R}}^*_{\widetilde{S},+}(\mathcal{F}^{kernel}) \leq r\Lambda \sqrt{\frac{1}{nK\tau_S}}, \tag{28}$$

$$\widehat{\mathfrak{R}}^*_{\widetilde{S},-}(\mathcal{F}^{kernel}) \leq r\Lambda \sqrt{\frac{1}{nK\tau_S}}. \tag{29}$$

*Proof.* For the first inequality, we can have

$$\widehat{\mathfrak{R}}^*_{\widetilde{S},+}(\mathcal{F}^{kernel}) = \mathbb{E}_{\boldsymbol{\sigma}} \left[ \frac{1}{m} \sum_{j \in [J]} \omega_j \sup_{\|\boldsymbol{w}\| \leq \Lambda} \left( \sum_{i \in I_j} \sigma_i \langle \boldsymbol{w}_{\alpha_i}, \Phi(\tilde{\boldsymbol{x}}_i^+) \rangle \right) \right]$$

$$\leq \mathbb{E}_{\boldsymbol{\sigma}} \left[ \frac{1}{m} \sum_{j \in [J]} \omega_j \sup_{\|\boldsymbol{w}\| \leq \Lambda} \|\boldsymbol{w}_{\alpha_i}\| \left\| \sum_{i \in I_j} \sigma_i \Phi(\tilde{\boldsymbol{x}}_i^+) \right\| \right] \quad \text{(Cauchy–Schwarz inequality)}$$

$$= \frac{\Lambda}{m} \sum_{j \in [J]} \omega_j \mathbb{E}_{\boldsymbol{\sigma}} \left[ \left\| \sum_{i \in I_j} \sigma_i \Phi(\tilde{\boldsymbol{x}}_i^+) \right\| \right] \quad \text{(the definition of the sup and linearity of expectation)}$$

$$\leq \frac{\Lambda}{m} \sum_{j \in [J]} \omega_j \left( \mathbb{E}_{\boldsymbol{\sigma}} \left[ \left\| \sum_{i \in I_j} \sigma_i \Phi(\tilde{\boldsymbol{x}}_i^+) \right\|^2 \right] \right)^{\frac{1}{2}} \quad \text{(Jensen's inequality)}$$

$$= \frac{\Lambda}{m} \sum_{j \in [J]} \omega_j \left( \mathbb{E}_{\boldsymbol{\sigma}} \left[ \sum_{p \in I_j, q \in I_j} \sigma_p \sigma_q \langle \Phi(\tilde{\boldsymbol{x}}_p^+), \Phi(\tilde{\boldsymbol{x}}_q^+) \rangle \right] \right)^{\frac{1}{2}}$$

$$= \frac{\Lambda}{m} \sum_{j \in [J]} \omega_j \left( \sum_{i \in I_j} \langle \Phi(\tilde{\boldsymbol{x}}_i^+), \Phi(\tilde{\boldsymbol{x}}_i^+) \rangle \right)^{\frac{1}{2}} \quad (\forall p \neq q, \mathbb{E}[\sigma_p \sigma_q] = 0 \text{ and } \mathbb{E}[\sigma_i \sigma_i] = 1)$$

$$\leq \frac{\Lambda r}{m} \sum_{j \in [J]} \omega_j \sqrt{m_j} \quad (\langle \Phi(\tilde{\boldsymbol{x}}_i^+), \Phi(\tilde{\boldsymbol{x}}_i^+) \rangle = \kappa(\tilde{\boldsymbol{x}}_i^+, \tilde{\boldsymbol{x}}_i^+) \leq r^2 \text{ and let } m_j = |I_j|)$$

$$= \frac{\Lambda r \chi_f(G)}{m} \sum_{j \in [J]} \frac{\omega_j}{\chi_f(G)} \sqrt{m_j}$$

$$\leq \frac{\Lambda r \sqrt{\chi_f(G)}}{m} \sqrt{\sum_{j \in [J]} \omega_j m_j} \quad (\sum_{j \in [J]} \frac{\omega_j}{\chi_f(G)} = 1 \text{ and Jensen's inequality}).$$

Since $\sum_{j \in [J]} \omega_j m_j = m, \chi_f(G) = \max\{|\mathbb{S}^+|, |\mathbb{S}^-|\}, m = |\mathbb{S}^+||\mathbb{S}^-|$ and $\min\{|\mathbb{S}^+|, |\mathbb{S}^-|\} = nK\tau_S$ hold, it comes

$$\widehat{\mathfrak{R}}^*_{\widetilde{S},+}(\mathcal{F}^{kernel}) \leq \frac{\Lambda r \sqrt{\chi_f(G)}}{m} \sqrt{\sum_{j \in [J]} \omega_j m_j} = \frac{\Lambda r}{\sqrt{nK}} \sqrt{\frac{1}{\tau_S}}.$$

Similarly to the proof of the first inequality, we can obtain the second inequality, which is omitted here for brevity. □

### B.5.1 PROOF OF COROLLARY 2

*Proof.* Based on Theorem 2 and Lemma B.5.1, it is straightforward to get the desired result. □

### B.5.2 PROOF OF COROLLARY 3

*Proof.* Based on Theorem 3 and Lemma B.5.1, it is straightforward to get the desired result. □

### B.5.3 PROOF OF COROLLARY 4

*Proof.* Based on Theorem 4 and Lemma B.5.1, it is straightforward to get the desired result. □

## C ADDITIONAL EXPERIMENTS

### C.1 ADDITIONAL EXPERIMENTAL SETUP

For all the experiments, we implement the algorithms with Pytorch (Paszke et al., 2019), where the stochastic gradient descent (SGD) is chosen as the optimizer, and the hyperparameter of the weight decay is selected in $\{10^{-6}, 10^{-5}, \ldots, 10^{-1}\}$ and we report the best Micro-AUC for each of the three algorithms. We use 3-fold cross-validation to get our average results for the experiments of linear and multilayer perception (MLP) models.

For the experiments of linear models on the tabular benchmark datasets, most datasets adopt a learning rate of 0.1 and converge at 200 epochs, except for the datasets of Image and enron, which have a learning rate of 0.01 and require 600 epochs to converge.

For the experiments of MLP models on the tabular benchmark datasets, we use a $3-$layer network, where the number of hidden layer's units is 100 for all benchmark datasets. The training epoch is set to 200 with a learning rate of 0.1 and batch size of 256.

For the experiments of CNN models on the (raw) image benchmark datasets, we use the test set in the original dataset as our test set and split the training set in the original dataset into train and validation sets according to the ratio of 7:3, where we select the hyperparameter of weight decay based on the validation sets. We use a batch size of 256 and train 40 epochs with a learning rate of 0.005 for both VGG-11 and ResNet-34.

### C.2 ADDITIONAL EXPERIMENTAL RESULTS

For the experiments of the linear model-based algorithms on the tabular benchmark datasets, we calculate the absolute upper bound values of these three algorithms, where the results are summarized in Table 6. Similarly to previous work (Wu et al., 2021; 2023), the absolute values might not reflect the true generalization error reasonably (i.e., bigger than 1), but we can still get valuable insights from the order of dependent quantities of these bounds.

Besides, we also conduct experiments with MLP-based learning algorithms on the tabular datasets, where the results are summarized in Table 7. From 7, we can observe that when the imbalance level of datasets is large (e.g., the last four datasets), $\mathcal{A}^{u_2}$ consistently performs better than $\mathcal{A}^{u_1}$. This corroborate our theory findings that $\mathcal{A}^{u_2}$ has an error bound of $O(\sqrt{\frac{1}{nK\tau_S}})$ while $\mathcal{A}^{u_1}$ depends on $O(\frac{1}{\tau_S}\sqrt{\frac{1}{nK\tau_S}})$. We also notice that when the imbalance level of datasets is small (e.g., the first five datasets), the algorithms perform comparably to each other. This is not inconsistent with our theory

Table 6: The mean upper bound values of three linear model-based algorithms for the tabular benchmark datasets. We set $\delta = 0.05$.

| Dataset | $\mathcal{A}^{pa}$ | $\mathcal{A}^{u_1}$ | $\mathcal{A}^{u_2}$ |
|---|---|---|---|
| CAL500 | 37.7 | 131.9 | 42.6 |
| emotions | 104.0 | 193.8 | 119.6 |
| image | 245.5 | 512.2 | 256.1 |
| scene | 127.5 | 397.0 | 138.1 |
| yeast | 24.4 | 48.3 | 29.9 |
| enron | 163.4 | 1362.8 | 170.6 |
| rcv1-s1 | - | 1044.3 | 60.3 |
| bibtex | - | 2683.2 | 81.4 |
| corel5k | - | 2364.6 | 45.1 |
| delicious | - | 603.5 | 24.0 |

Table 7: Micro-AUC (mean $\pm$ std, the symbol . means 0.) of all three MLP model-based algorithms on tabular benchmark datasets. The top two algorithms on each dataset are highlighted in bold and the top one is labeled with $\dagger$. Besides, "-" means that $\mathcal{A}^{pa}$ takes more than one week by using a GPU server on the corresponding datasets.

| Dataset | $\mathcal{A}^{pa}$ | $\mathcal{A}^{u_1}$ | $\mathcal{A}^{u_2}$ |
|---|---|---|---|
| CAL500 | $.8111 \pm .0034$ | $.8022 \pm .0016$ | $\mathbf{.8151} \pm \mathbf{.0016}^{\dagger}$ |
| emotions | $\mathbf{.8736} \pm \mathbf{.0138}$ | $\mathbf{.8740} \pm \mathbf{.0146}^{\dagger}$ | $.8658 \pm .0130$ |
| image | $\mathbf{.8818} \pm \mathbf{.0090}^{\dagger}$ | $\mathbf{.8741} \pm \mathbf{.0082}$ | $.8723 \pm .0049$ |
| scene | $.9530 \pm .0040$ | $\mathbf{.9535} \pm \mathbf{.0037}$ | $\mathbf{.9560} \pm \mathbf{.0035}^{\dagger}$ |
| yeast | $\mathbf{.8478} \pm \mathbf{.0014}$ | $.8390 \pm .0012$ | $\mathbf{.8497} \pm \mathbf{.0014}^{\dagger}$ |
| enron | $.9025 \pm .0112$ | $\mathbf{.9132} \pm \mathbf{.0020}$ | $\mathbf{.9219} \pm \mathbf{.0040}^{\dagger}$ |
| rcv1-s1 | - | $.9593 \pm .0015$ | $\mathbf{.9632} \pm \mathbf{.0012}^{\dagger}$ |
| bibtex | - | $.9373 \pm .0010$ | $\mathbf{.9431} \pm \mathbf{.0005}^{\dagger}$ |
| corel5k | - | $.8594 \pm .0037$ | $\mathbf{.8881} \pm \mathbf{.0042}^{\dagger}$ |
| delicious | - | $.8757 \pm .0011$ | $\mathbf{.9131} \pm \mathbf{.0008}^{\dagger}$ |

findings of the dependent order of the imbalance level on these bounds. When the imbalance level is small (or equivalently, the dataset is nearly balanced), the bounds of these algorithms are close to the same order and thus it is hard to see which is better. Notably, in this case, we also notice that the relative performance is also dependent on the choice of the hypothesis space by comparing the results of the linear (Table 3) and MLP model (Table 7), which needs further investigation, left as future work.

Additionally, we also conduct experiments with another popular CNN-based model (i.e., VGG-11) on the image benchmark datasets, where the results are summarized in Table 8. The empirical results also confirm our theory findings.

Table 8: Micro-AUC of all three algorithms with the VGG-11 model on image benchmark datasets. The top two algorithms on each dataset are highlighted in bold and the top one is labeled with $\dagger$. Besides, "-" means that $\mathcal{A}^{pa}$ takes more than one week by using a GPU server on the corresponding datasets.

| Dataset | $\mathcal{A}^{pa}$ | $\mathcal{A}^{u_1}$ | $\mathcal{A}^{u_2}$ |
|---|---|---|---|
| PASCAL VOC 2012 | $\mathbf{.8125}$ | .6759 | $\mathbf{.8225}^{\dagger}$ |
| MSCOCO 2017 | - | .7303 | $\mathbf{.9004}^{\dagger}$ |
| NUS-WIDE | - | .8587 | $\mathbf{.9385}^{\dagger}$ |

