# OpenReview forum: "Matrix-wise Class Imbalance Matters: On the Generalization of Micro-AUC in Multi-label Learning"
_ICLR.cc/2024/Conference — ICLR 2024 Conference Withdrawn Submission_

### Official Review · Reviewer_dXyR · 2023-10-25

**Soundness:** 4 excellent
**Presentation:** 4 excellent
**Contribution:** 3 good
**Rating:** 8
**Confidence:** 3

**Summary:**

This paper intends to give novel insights for Micro-AUC optimization in multi-label classification. To this end, three types of surrogate losses are theoretically analyzed: (1) a classical univariate loss commonly used in MLC, (2) a pairwise loss commonly used in AUC optimization, and (3) a reweighted univariate loss that is in accordance with oversampling the minority class.

The authors derive generalization bounds for the three types of loss functions. To this end, a new quantity is introduced: the matrix-wise class imbalance of an MLC dataset. The results show that the traditional univariate loss is inferior to the other two losses. The last two losses are competitive, so each of them could be good choices to train an MLC classifier. In addition, the authors also present some experimental results that confirm the theoretical results.

**Strengths:**

- This paper is well written. The authors do a good job in focussing on the key results, while moving the technical derivations to an appendix.
- As far as I know, this analysis is novel.
- The experiments seem to confirm the theoretical results.

**Weaknesses:**

I didn't spot any big weaknesses in this paper. However, I have to admit that I am not a "bound prover", so I was unable to check the correctness of the proofs.

Some aspects in which the paper could be further improved are:
1. The key findings look very similar to the following paper: W. Kotlowski et al. Bipartite Ranking through Minimization of Univariate Loss, ICML 2011. That paper is not discussed. What's novel compared to the findings in that paper?

2. There are three ways of doing averaging for AUC and other multivariate loss functions: micro-averaging, macro-averaging and instance-wise averaging. Of those three, I have always felt that micro-averaging is the least important in applications. Are there actually applications where this form of averaging is relevant? I cannot think of such an application.

3. Are the results immediately extendable to the other types of averaging?

4. Micro-, macro and instance-wise averaging are also used in other multi-target prediction settings, such as dyadic prediction and matrix completion. Are the results immediately extendable to such settings?

5. Is a similar type of analysis possible for other multivariate loss functions, such as F-measure, PR-AUC, Jaccard, etc.?

I do understand that the authors cannot answer all these questions in a single paper, but extending the related work section in that direction would be useful.

Small remark on the notation: The definition of the sets S^+, S^-, S^+_i, S^-,i looks a bit awkward. I think that you can simply write S^+ = {(p,i) | y_pi = 1} etc.

**Questions:**

See above.

---

### Official Review · Reviewer_MN9U · 2023-10-30

**Soundness:** 3 good
**Presentation:** 3 good
**Contribution:** 2 fair
**Rating:** 5
**Confidence:** 4

**Summary:**

This paper defines a new data-dependent quantity to characterize the degree of class balance in the data matrix and further analyzes the generalization bound of different alternative loss functions.

This paper also proposes a reweighting univariate loss and proves its effectiveness through the aforementioned bound.

**Strengths:**

S1. The writing is clear and concise, making it easy to understand both theory and methods.

S2. The theoretical proof in this paper relies on the Bipartite Ranking (BR) (Usunier et al., 2005; Amini & Usunier, 2015), which is a relatively new theoretical tool in the MLL field. It has been used in [Wu et al., 2023] to prove the generalization bound.

**Weaknesses:**

W1. My opinion is that the motivation of the research is not clear enough. Considering that related research fields have the latest results[Wu et al. 2023], the paper does not provide a detailed comparison of the theoretical results of these two works. At least in my judgment, the theoretical results of the two papers are very similar. The order of bound is the same, and the main difference is only in the definition of class imbalance.

W2. The experimental verification of this paper is not enough. In comparison with the simple demonstration of the performance of each dataset, I believe that the demonstration of the relationship between the new data-dependent quantity and generalization performance is more important. Especially as a theoretical work, it is necessary to verify the real effect of the critical factor $\tau$ in the bound.

W3. I noticed that the paper has a high similarity with [Wu et al. 2023] in terms of writing and research problem, but it did not provide a detailed comparison.

**Questions:**

Q1. The similarity between this paper and [Wu et al. 2023] in the abstract and introduction sections is extremely high, and I believe this issue is very serious. Although the two have different definitions of class balance, this writing style may leave a negative impression on the reviewer. I would like to ask what changes have occurred in the research questions and methods from [Wu et al. 2023] to the work of this paper.

Q2. Referring to W2 above, may I ask if there is a significant difference in the theoretical results between these two papers? What are the advantages or innovations of this paper in terms of heuristic algorithm design?

**Details Of Ethics Concerns:**

Not very serious, but with high similarity to [Wu et al. 2023]'s writing.

---

### Official Review · Reviewer_yuay · 2023-10-31

**Soundness:** 3 good
**Presentation:** 3 good
**Contribution:** 2 fair
**Rating:** 5
**Confidence:** 4

**Summary:**

This paper focuses on characterizing the generalization guarantees of the Micro-AUC measure in multi-label learning. The authors find that the matrix-wise class imbalance affects the generalization bounds. Besides, the authors design a reweighted univariate loss and verify its effectiveness by empirical experiments on various benchmarks.

**Strengths:**

1. This paper is well-written, the clarity and coherence of the writing style make it easy to follow.
2. This paper provides solid theoretical results with respect to the Micro-AUC measure.

**Weaknesses:**

1. It is important to point out that this paper bears a striking resemblance to reference [1]. Except for the different measures (Micro-AUC vs Macro-AUC in [1]), different imbalance quantities (matrix-wise class imbalance vs label-wise class imbalance in [1]), the techniques, the fractional Rademacher complexity, and the main results are very similar to [1]. Given so many similarities, it is necessary to critically assess the novelty and contribution of this paper, as it is quite limited for me.
2. The experimental evaluation of this paper focuses solely on the three proposed losses, without comparing them against commonly used baseline losses, such as BCE (Binary Cross Entropy) loss, focal loss[2], and asymmetric loss[3]. This omission raises doubts about the practical effectiveness of the proposed losses.

[1]Towards understanding generalization of macro-auc in multi-label learning. ICML 2023.

[2]Focal loss for dense object detection. ICCV 2017.

[3]Asymmetric loss for multi-label classification ICCV 2021.

**Questions:**

1. $\bf Compared \space with \space [1]$, what are the $\bf{new}$ challenges for providing learning guarantees of the Micro-AUC measure, and what are the most significant $\bf{technical}$ differences?
2. On page 3, Eq. (4), the authors take the logistic loss $ℓ(t) = log_2(1 + exp(−t))$ as an example for the base loss function ℓ(t), however, this loss is unbounded, which violates the third condition of Assumption 1.

[1] Towards understanding generalization of macro-auc in multi-label learning. ICML 2023.

---

### Official Review · Reviewer_wJZ6 · 2023-10-31

**Soundness:** 3 good
**Presentation:** 3 good
**Contribution:** 3 good
**Rating:** 8
**Confidence:** 2

**Summary:**

This paper analyses the generalization performance of the Micro-AUC with three surrogate losses. The proposed theorems show that the univariate loss has a worse upper bound of the generalization gap than pair-wise and reweighting univariate losses. Finally, empirical results verify the theory findings.

**Strengths:**

This paper is clear and easy to follow. The proposed theorems fill the gap of the theoretical analysis of Micro-AUC.

**Weaknesses:**

There are some specialized terms in graph theory, e.g., fractional independent vertex cover, fractional chromatic number, which appear without definitions. It would be better to add them in preliminaries or appendix.

**Questions:**

I wonder what the difference is between the fractional Rademacher complexity and the traditional Rademacher complexity. Is there an intuitive explanation of the fractional Rademacher complexity.

---

### Official Review · Reviewer_6xUb · 2023-11-01

**Soundness:** 3 good
**Presentation:** 3 good
**Contribution:** 2 fair
**Rating:** 6
**Confidence:** 3

**Summary:**

To fill up the gap of theoretical understanding of Micro-AUC, this paper characterizes the generalization guarantees of algorithms based on three surrogate losses w.r.t. Micro-AUC. This paper identifies the matrix-wise class imbalance and shows that the commonly-used univariate loss-based algorithm has a worse learning guarantee than the ones with the proposed pairwise and reweighting univariate loss theoretically. Moreover, the experiments are consistent with their theory.

**Strengths:**

1. This paper is sound and clearly written.
2. This paper derives learning guarantee for learning algorithm for three different surrogate losses.

**Weaknesses:**

1. The contribution lacks enough novelty. This paper proposes two new surrogate losses, but the natural surrogate pairwise loss takes too much time to converge from the experiments. The reweighted univariate loss is also based on the widely-used ordinary univariate loss.
2. The Related Work is the 6th session, which is not common. I am wondering if there is any specific reason for putting Related Work there.

**Questions:**

1. In Definition 1, $\tau_S \in [\frac{1}{K},\frac{1}{2}]$, to guarantee that $\tau_S \geq \frac{1}{K}$, do we need to assume that for the label of each data point $y_i$, we need at least one +1 or -1?